# Electron-Beam-Pumped UVC Emitters Based on an (Al,Ga)N Material System

**DOI:** 10.3390/nano13142080

**Published:** 2023-07-15

**Authors:** Valentin Jmerik, Vladimir Kozlovsky, Xinqiang Wang

**Affiliations:** 1Ioffe Institute, 26 Politekhnicheskaya, 194021 St. Petersburg, Russia; 2P. N. Lebedev Physical Institute, Leninsky Ave. 53, 119991 Moscow, Russia; kozlovskiyvi@lebedev.ru; 3State Key Laboratory for Mesoscopic Physics and Frontiers Science Center for Nanooptoelectronics, School of Physics, Peking University, Beijing 100871, China; wangshi@pku.edu.cn

**Keywords:** ultraviolet C emission, electron-beam-pumped ultraviolet (UV)C emitters, AlGaN group III-nitrides, low dimensional structures, 2D quantum wells

## Abstract

Powerful emitters of ultraviolet C (UVC) light in the wavelength range of 230–280 nm are necessary for the development of effective and safe optical disinfection technologies, highly sensitive optical spectroscopy and non-line-of-sight optical communication. This review considers UVC emitters with electron-beam pumping of heterostructures with quantum wells in an (Al,Ga)N material system. The important advantages of these emitters are the absence of the critical problem of p-type doping and the possibility of achieving record (up to several tens of watts for peak values) output optical power values in the UVC range. The review consistently considers about a decade of world experience in the implementation of various UV emitters with various types of thermionic, field-emission, and plasma-cathode electron guns (sources) used to excite various designs of active (light-emitting) regions in heterostructures with quantum wells of Al_x_Ga_1−x_N/Al_y_Ga_1−y_N (x = 0–0.5, y = 0.6–1), fabricated either by metal-organic chemical vapor deposition or by plasma-activated molecular beam epitaxy. Special attention is paid to the production of heterostructures with multiple quantum wells/two-dimensional (2D) quantum disks of GaN/AlN with a monolayer’s (1 ML~0.25 nm) thickness, which ensures a high internal quantum efficiency of radiative recombination in the UVC range, low elastic stresses in heterostructures, and high-output UVC-optical powers.

## 1. Introduction

Optoelectronic devices based on wide-bandgap compounds in an (Al,Ga)N material system are capable of operating in a significant part of the ultraviolet C (UVC) spectral range with a wavelength (λ) between 210 and 280 nm. Many UVC photodetectors and emitters have been developed over the last two decades due to the many applications of these devices, reviewed in the next chapter. UVC light-emitting diodes (UVC-LEDs) have received the most attention from UVC emitter developers and many excellent books, reviews and roadmaps on the subject have been published in recent years [1,2,3,4,5,6,7,8]. A brief analysis of the literature reveals the limited use of AlGaN-based semiconductor UVC emitters in the main practical applications. Today, although AlGaN-based UVC-LEDs have achieved a record wall-plug efficiency (WPE) of 20% [9], which is close to the ~30% efficiency for Hg lamps emitting at λ = 253.7 nm, typical values of this parameter for commercial devices lie in the 1–5% range. UVC-LEDs are just beginning to penetrate the UVC optoelectronic market [10].

The relatively low WPE of UVC LEDs limits their output optical power to no more than ~100 mW at 250 mA in commercial devices. However, according to the forecast published by Kneissl et al. in 2019 [11], this parameter for the UVC-LEDs emitting in the range λ = 256–310 nm should be increased to 15–20% by 2025, which is quite close to the efficiency of Hg lamps. This progress, combined with the environmental, space and other advantages of semiconductor devices, will ensure their wider practical application. Meanwhile, the further progress for far-UVC-LEDs based on AlGaN heterostructures is not so optimistic, and today the best laboratory device of this type, emitting at 231 nm, demonstrates a maximum cw- optical power of ~3.5 mW at 150 mA [12]. This decrease in far-UVC-LED power at shorter operating wavelengths is largely due to the increased difficulty of doping *p*-emitters with a higher Al content in such devices. Most researchers suppose this problem to be the most difficult challenge in this area [13,14,15,16,17].

Therefore, it is of interest to develop alternative UVC emitters based on AlGaN heterostructures without *p*-type-doped AlGaN layers, in which luminescence is provided by pumping with an electron (e-) beam with an energy of up to 20 keV and a current from several milliamps in continuous wave (cw-) mode to a few amperes in a pulsed one. In addition to eliminating the need for the doping of these structures, they have other advantages, such as design simplicity and large emitting surface areas up to ~20 cm^2^ without any current leakage issue.

In this review, first, a brief review of the most promising areas of application of e-beam-pumped UVC emitters is carried out. Then, the main types and design features of electron sources (e-guns) used for the e-beam pumping of semiconductor light-emitting AlGaN-based heterostructures are considered. Specific information is given about various e-guns that provide the emission of e-beams in an extremely wide current range from tens of nA to several amperes, as well as having different beam diameters from the submillimeter range to two inches. In the main chapter of the review, we successively consider UVC emitters with various designs of active (light-emitting) regions, which, as a rule, are based on heterostructures with multiple quantum wells (MQW) in an (Al,Ga)N system, excited by various e-guns operating in cw- and pulsed mode. The possibilities of fabrication of UVC emitters with various beam sizes and an output optical power from several milliwatts to several tens of watts are demonstrated. Special attention is paid to a new type of heterostructures based on monolayer (ML)-thick GaN/AlN MQWs.

## 2. Main Applications of UVC Emitters

The main applications of conventional lamp-based and new semiconductor UVC emitters are schematically shown in Figure 1. By far, optical (chemical-free) disinfection of air, water and surfaces is the most common and most demanded application for UVC radiation. The earliest scientific observations of the bactericidal effects of sunlight radiation began with Downes and Blunt in 1877 [18], and its practical application was accelerated by the simultaneous development of the first mercury (Hg) vapor arc lamps. In 1903, Barnard and Morgan discovered the germicidal effect of UV light in the spectral range of 226–328 nm [19]. These UVC-emitting lamps were first applied in 1906 for drinking water treatment [20]. Since the 1930s, these lamps have been widely introduced into hospital and school practice for the optical disinfection of rooms, instruments, etc. A detailed history of the discovery and introduction of UV radiation for optical disinfection can be found in Kowalski’s book [21]. The discovery of the structure of DNA molecules in the 1950s scientifically explained the biological activity of UVC light. This effect was associated with various optically induced rearrangements of bonds in DNA molecules included in all pathogenic viruses and bacteria, which prevented their reproduction and activity.

Currently, in the context of the just-ended SARS COVID19 pandemic and possible future pandemics of no less dangerous diseases, the optical disinfection of air/water and surfaces is being actively developed by many companies and research centers all over the world. The state of the art in this area has been described in excellent recent reviews [22,23,24], which indicate that exposure to UVC radiation at 254 or 222 nm with a dose of about 2–6 mJ·cm^−2^ is required to reduce the concentrations of the main pathogenic bacteria by a thousand times. The same UVC radiation can also be used to protect food from bacteriological decay [25,26].

Due to high biological activity, UV light has several harmful side effects, namely a carcinogenic effect on human skin and the development of eye cataracts [27]. The maximum risk of the former is manifested in the so-called UVB subrange (λ = 280–315 nm), and data on the photocarcinogenic effect for UVC-GI (germicidal irradiation) range (λ = 250–280 nm) are still contradictory and insufficient [28]. It is well-established that UV light with λ = 250–315 nm can penetrate down to the basal cell layer, the bottom-most layer of the skin, and damage human DNA. However, these effects disappear almost completely in the so-called far-UVC range (λ = 200–230 nm), which has a similar biological effect on pathogenic µm size microorganisms, but does not penetrate into the human stratum corneum (the outer dead layer of the skin with a thickness of 5–20 µm), nor into the cornea of the eye (thickness of ~500 µm) [29,30,31]. According to the 8 h threshold limit values for corneal and skin UV damages established in 2022 by the American Conference of Governmental Industrial Hygienists, the minimum safe dose for UV light with λ = 250–300 nm is 10 mJ·cm^−2^, and for UVC light at 222 nm, it increases to 5 × 10^2^ mJ·cm^−2^ [32]. Thus, the UVC range is divided into two subranges: high-risk UVC-GI (λ = 250–280 nm) and safe far-UVC (λ = 200–230 nm).

Currently, the main means of optical disinfection are mercury (Hg) lamps, emitting mainly in the UVC-GI subrange at λ = 253.7 nm. However, recently, these lamps have been intensively replaced by excimer (Kr–Cl) lamps of a new type, which emit mainly in the far-UVC subrange at λ = 222 nm. These new lamps have a similar biological activity, but are much safer for humans and therefore are considered more promising.

The main advantage of the above lamps is the relatively high WPE up to 35%, providing output UVC optical powers from a few watts to several tens of watts at typical optical power density of ~1 mW·cm^−2^ and typical distances from UVC lamps of ~10–20 cm [21,31,33,34,35,36]. Thus, the required lower limit of the UVC radiation dose of ~10 mJ·cm^−2^, necessary for the full destruction of the main pathogens, can be quickly achieved to ensure the disinfection of running water, air in enclosed spaces (rooms) and various large-area surfaces.

However, UVC-GI lamps have disadvantages and, first, low environmental friendliness, due to the presence of mercury in these lamps, the use of which is banned under the UN Minamata Convention on Mercury [37]. Second, it is necessary to filter parasitic UVC radiation in the ozone-generating subrange of λ < 185 nm, which increases with an increasing lamp power of any type. Unfortunately, this problem has not been completely solved in today’s lamps, as evidenced by studies of the effect of a side increase in the concentration of ozone and numerous types of nanoparticles in rooms during the operation of high-power UVC-GI lamps [33]. Moreover, for all types of UVC lamps, in addition to their fragility, large dimensions, high supply voltage, long warm-up times and short service life, it is also necessary to note their principal limitations, such as a limited set of operating wavelengths and the impossibility of generating UVC laser emission.

Another application of UVC irradiation is optical spectroscopy with the emission of exciting UVC light and detection of a fluorescence signal in the UVB, near UV and visible ranges. These methods are used to identify various substances, including various hazardous bio and chemical agents, explosive substances [38,39,40], drugs [41,42], molecular assemblies, such as peptides/proteins [43], as well as the traces of water and organic molecules in astrobiology research, including the study of Mars [44,45], etc. It is especially important to ensure the development of UVC emitters operating in the range of λ = 200–250 nm, where strong fluorescence cross sections improve the detection sensitivity. Various types of gas-discharge lamps, including low-pressure Hg, deuterium, halide, etc. (see Figure 1), are used in UV-spectroscopic devices to excite the recorded spectra.

A much higher accuracy is ensured in the case of laser-induced fluorescence (LIF) detection using various types of UVC lasers, including solid-state lasers (Nd3^+^-YAG: 4th and 5th harmonics at 266 and 213 nm, respectively [38]), metal vapor lasers (Ne–Cu: 248.6 nm, He–Ag: 224.3 nm), etc. [40]. Moreover, the use of UVC lasers makes it possible to realize the fusion of Raman fluorescence detection with fluorescence-free Raman detection to improve the sensitivity of the method [41,43,44,45]. To improve the accuracy of UVC optical spectroscopy methods, including light detection and ranging (LIDAR), it is necessary to provide the ability to generate UVC radiation in the entire wavelength range with maximum power [38]. Compact, efficient, durable, high-power UVC lasers would greatly advance the LIF detection of environmental and biological threats. At present, several scientific groups are engaged in the development of alternative semiconductor UVC lasers based on AlGaN heterostructures, but this topic is beyond the scope of this review. A detailed analysis of the challenges and implementation of efficient electrically driven UVC lasing can be found in a recent review by Nikishin et al. [46].

In addition, UVC radiation with λ = 200–280 nm can be used to implement non-line-of-sight (NLOS) optical communication at distances from several tens of meters to one kilometer or more [47,48,49]. This unique ability is due to the strong Mie scattering of UVC radiation by the air atmosphere/aerosol cloud and solar blind nature of this radiation. The choice of operating UVC wavelength is determined by a trade-off between increased Mie scattering and a shorter communication range in the shorter wavelength range. Various types of UVC emitters are proposed to be used as transmitters in NLOS systems. They include both UVC lasers emitting coherent radiation and various sources of spontaneous radiation—mercury lamps or semiconductor UVC LEDs. The most important parameter that determines the communication range and data transfer rate is the output power of the emitters. For example, various solid-state UVC lasers with a frequency multiplier or a nonlinear crystal converter of the initial visible or IR laser emission are used as such transmitters in the NLOS test [48]. An excellent review on the use of various UVC emitters in NLOS systems, including various mercury lamps, solid-state lasers, and the latest UVC-LEDs, was recently published by Guo et al. [49].

Research and development on applications of semiconductor e-beam-pumped UV emitters in the areas described above are at an early stage. Thus, Yoo et al. [50] described the successful deactivation of *E. coli* bacteria with 5-log reduction for 60 s using a sapphire-based UVC emitter at a peak λ = 230 nm with a light-emitting area of 8.03 cm^2^ and a total optical output power of 88.65 mW. This emitter was pumped by a carbon nanotube (CNT) e-gun with an operating voltage of 10 kV and an anode current of 0.7 mA. DeFreez et al. have developed new e-beam-pumped AlGaN-based UVC emitters operating in the wavelength range from ~220 nm to detect biological and chemical threats [51]. The test emitters demonstrated in this work showed much-higher-output optical powers compared to longer-wavelength AlGaN-based UVC LEDs emitting above 255 nm (unfortunately, in this work, the powers were measured only in relative units). Recently, Kang et al. [52] demonstrated a compact light tube for NLOS applications in which a UVC emitter made from a YPO_4_:Bi^3+^ layer provided cw- emission at 246 nm with an output optical power of up to 400 mW when excited by a thermionic e-gun with an e-beam energy of 10 keV and an anode current up to 0.8 mA.

## 3. General Principles of Electron-Pumped UV Light Emitters

The main problems of creating a wide class of e-beam-pumped light-emitting devices are considered in a recent review by Cuesta et al. [53]. First of all, e-beam-pumped UVC emitters (e-beam tubes) are divided according to radiation output geometry into two types, schematically shown in Figure 2. In the first (I) type of such e-beam tubes, shown in Figure 2a,b, UVC radiation is emitted through a surface that is bombarded by electrons from an e-gun cathode located above this surface [54,55]. Such a design can be implemented using e-beam-emitting cathodes in the form of either a grid or segmented cathodes that provide oblique beam propagation. The main advantage of these e-beam tubes is the possibility of organizing effective heat removal through the substrate and the absence of requirements for its transparency.

However, in practice, light emitters of a second type (II) with another e-beam pumping scheme are most widely used [56,57,58]. In these emitters, shown in Figure 2c–e, the e-beam also bombards the surface with the active (light-emitting) region, but the output radiation is emitted through the opposite rear substrate surface. Figure 2c shows the scheme of excitation of an Al_x_Ga_1−x_N/Al_y_Ga_1−y_N (*y* > *x*) MQW heterostructure in a vacuum chamber using an e-gun that emits an e-beam with controlled energy, current and various operating modes—continuous wave (cw-) or pulsed [56]. The heterostructure is placed on a quartz vacuum window, through which UVC radiation exits to measure its characteristics using a photocathode and a spectrograph. Several types of e-guns can be used in one installation (a detailed description of the various types of e-guns will be given in the next chapter). During measurements, it is important to ensure the stability of the properties of the surface bombarded by an e-beam with an energy of up to 20–30 keV. In addition, it is necessary to exclude the charging of the heterostructure dielectric surface under the action of this e-beam. Both tasks are quite successfully solved by sputtering a thin (0.2–0.7 µm) metal (Al) film onto the bombarded surface.

A compact design of an e-beam-pumped light emitter of type II is shown in Figure 2b, where all parts are included in the standard lamp bulb R63 with the bulb comprising a built-in AC–DC high-voltage converter (up to 20 kV) of the electric line voltage (50 Hz, 230 V) [57]. Figure 2d shows another type II UVC emitter excited by an e-gun with a large-area cathode based on carbon nanotubes (CNTs) [58].

The total thickness and position of the active region in the AlGaN-based e-beam-pumped UVC emitters are determined by the spatial distributions of the deposited e-beam energy in the AlGaN layers, which is usually calculated using the standard CASINO program [59]. Figure 3 shows the results of these calculations for various initial e-beam energies [60].

## 4. The Main Types of Electron Guns Used to Excite UVC Radiation

A reliable electron source (e-gun) for the excitation of a specimen with an optically active (Al,Ga)N-based heterostructure is one of the most important parts of any e-beam-pumped UVC emitter. Figure 4 classifies the main types of these e-guns with different basic principles of electron emission, operating parameters and output characteristics. Among the latter, the most important characteristic is the e-beam current, which primarily determines the output optical power of the UV emitter. This chapter describes successively the e-guns that have already found use in UVC emitters and can do so. The specific parameters and characteristics of these e-guns are summarized in Table 1 at the end of this chapter.

### 4.1. Thermionic e-Beam Emission Guns

The first type of e-beam source in Figure 4 is called thermionic, since electron emission is provided by heating the incandescent cathode. In practice, the several types of thermionic e-guns differ primarily in the design of the cathode assembly. Figure 5 shows the typical designs of e-guns with thermionic (incandescent) cathodes, usually made of a V-shaped tungsten filament with a work function (W) of 4.5 eV (thoriated tungsten has a work function of 2.7–4.5 eV) or one of the crystals of rare-earth metal hexaborides (LaB_6_, CeB_6_) with a much lower work function of about 2.4–2.7 eV for LaB_6_ [61,62,63]. At sufficiently high temperatures of the cathode material (1400–2400 °C), a certain percentage of electrons acquires the energy sufficient to exceed the work function, and these electrons are emitted in accordance with Richardson’s law, which relates the current density from the e-gun (*J*) to the operating temperature (*T*) in Kelvin.
(1)$$J=A\cdot T^{2}e^{-\frac{W}{kT}},$$

Here *k* is the Boltzmann constant (8.6 × 10^−5^ eV·K^−1^) and *A* is Richardson’s constant (A·m^−2^K^−2^), which is only constant for a given cathode material. The electrons leaving the filament are accelerated by a potential difference, in the typical range from a few hundred Volts to 30 kV, between the filament and an anode. This voltage determines the energy of the electrons in the beam (*E_e_*) bombarding the surface of the light-emitting structure in the UV emitter. The filament is surrounded by a grid cap, the Wehnelt cylinder, biased at a negative potential (0–2.5 kV). The electric field generated by the filament, the Wehnelt cylinder, and the anode causes the emitted electrons to converge to a crossover point with a diameter of 10–50 μm and a certain divergence angle below the cylinder. An electromagnetic coil is used to focus the e-beam into a spot whose diameter on the light-emitting surface varies from <10 µm up to ~1 mm.

Pure or thoriated tungsten cathodes provide stable e-emission under normal vacuum conditions (~10^−5^ mbar) with an extended filament lifetime at high temperatures. The typical e-beam current (I_e_) varies from several tens of nanoamperes to a few milliamps [60]. Higher e-beam currents up to several tens of mA can be obtained using a thermionic cathode made of lanthanum hexaboride (LaB_6_) [64,65]. Although heating the thermionic source gives a higher e-beam current, higher temperatures shorten the source life through evaporation and/or oxidation. Therefore, a compromise operating temperature is chosen. These e-guns are widely used in the analytical study of cathodoluminescence (CL) spectra, electron microscopy, television, medicine, etc. The main disadvantage of thermionic e-guns, except for their relatively low currents, is the limited size of the output beams (~1 mm), which leads to the need to spatially scan the e-beam by alternating electric fields over the surface to expand the radiation pattern and to improve the dissipation of thermal power in the light-emitting structures.

### 4.2. Field Emission (Cold) Electron Sources

Field emission (cold) electron sources operate in a fundamentally different way to a thermionic e-beam gun [61]. The basic principle of these sources is that the electric field strength (*E*) increases significantly at sharp points, since the magnitude of the strength for the voltage (*V*) applied to a spherical point of radius (*r*) is calculated as
(2)$$E=\frac{V}{r}$$

Currently, many technologies allow one to fabricate fine needle “tips”. One of the easiest materials to produce an array of nano-sized tips with a radius of <100 nm is carbon. If we apply a 1 kV potential to this tip, then *E* is ~10^10^ Vm^−1^ and this lowers the work-function barrier sufficiently for electrons to tunnel out of the carbon.

One of the first UVC emitters with a carbon (graphene)-based field e-emitter was developed by the Matsumoto group in 2012 [66]. Figure 6a shows a nanotip fabricated by etching carbon nanorods in hydrogen plasma, which led to the formation of an array of nanotips with a radius at the top of 5 nm, having a crystallographic structure corresponding to graphene. A stable behavior of this cathode under a current of 0.5 mA and e-beam energy of 15 keV for 5000 h was demonstrated, as shown in Figure 6b. This e-emitter was evacuated to a base pressure of 10^−7^ Pa, and then, the glass tube was chipped off.

The compact size of this e-gun makes it possible to combine it in one housing (30 mm in diameter and 60 mm long) with a UVC-emitting heterostructure grown on a sapphire substrate, which is shown in Figure 5c. It should be noted that the power supply of the e-beam tube is also compact and has dimensions of 100 × 150 × 35 mm and a weight of 650 g. Because of its low current consumption during operation, the device can be handheld and driven by AA dry batteries.

Several groups are currently intensively developing various types of large-area e-sources using field emission from carbon nanotubes (CNTs) distributed over the surface of various substrates (Si, stainless steel) up to two inches in diameter [50,55,58,67,68,69,70,71]. As a rule, these emitters use a triode electrode scheme, in which a lower voltage (1–2 kV) is applied to the first metal mesh electrode (gate) located close (less than 1mm) to the surface of the e-emitting cathode, as shown in Figure 7. This voltage determines the output of the electrons with characteristic currents of 1–5 mA. A higher voltage (up to 7 kV) applied to the second far electrode (anode) determines the energy of the electrons bombarding the light-emitting structure located on this electrode. Most of the electrons emitted from the CNT cathode reach the anode at a typical distance between these electrodes of about 20 mm [71].

There are several approaches to the fabrication of CNT arrays on the substrates. In the first of these, illustrated in the inset to Figure 7a, the regularly arranged positions of individual nanotip CNTs are controlled usually using conventional photolithography [50,58,67,68,70]. The nanotips are formed using plasma-enhanced chemical vapor deposition (CVD) in location with a selectively fabricated Ni catalyst on Si substrates. CNT emitters fabricated in this way have a typical height of 40 µm, an apex diameter of 50–100 nm and a bottom contact diameter of a few microns. The individual CNTs are arranged periodically at 30 µm intervals in large islands (dots) consisting of 49 CNT emitters. The islands are patterned at 0.5 mm intervals to decrease cathode current leakage through the gate electrode. CNT e-emitters of this type have a typical e-emitting area of 200–300 mm^2^. Figure 8a shows typical current–voltage characteristics for all currents in the CNT triode e-emitter, measured in continuous wave (cw-) mode, and its photo (Figure 8b) with an output fluorescent screen emitting visible light under UVC excitation [50].

Alternatively, CNT e-beam emitters may be fabricated using a complex multi-step process in which multi-walled CNT powder is first prepared using thermal CVD [55,69,71]. The CNTs are characterized by high crystallographic quality and have an outer diameter of 7 nm, approximately five walls and a length of more than 10 µm. CNT paste is then synthesized by mixing the CNT powder with various organic and non-organic solvents and fillers. The CNT paste is spread over a stainless-steel substrate using a simple screen-printing process to form the desired arrays of CNT dots with a diameter of about 300 µm. The CNTs dots are consequently thermally treated in air and vacuum ambients to remove all organic ingredients for the stabilization of the e-emissive characteristics. Finally, an adhesive tape and soft roller are employed for the vertical alignment of the nanotubes, providing their physical activation. This type of CNT e-emitter ensures a maximum output e-current of up to 5.5 mA in cw- mode and a largest emitting surface diameter of up to 2 inches, as shown in Figure 9a [71].

Comparing the two methods of CNT fabrication employing either direct selective growth using plasma-enhanced CVD or screen printing of CNT paste, one can highlight the positive and negative sides of each method. The first method makes it possible to directly obtain vertically oriented CNTs with a precisely defined arrangement of individual nanotips and their uniform distribution over the area of the substrate. The main advantage of the second method is its high cost efficiency and the high productivity of the process on large-diameter substrates. However, it should be considered that CNTs fabricated in this way are much more poorly vertically aligned than nanotips obtained by the plasma-enhanced CVD, and to improve their alignment, it is necessary to perform additional technological operations of annealing, aging, etc.

In summary, it should be noted that all CNT e-emitters have the smallest dimensions at the largest e-emitting area compared to other types of e-guns (sources), as well as zero warm-up time, operation at room temperature and relatively low operating voltages and currents. The CNT e-emitter can operate in a high vacuum of 10^−7^ Torr. These devices do not have high-temperature parts and do not consume a high heating current, as in thermionic e-emitters. Unfortunately, very little is known about the temporal stability of CNT electron sources, and this issue needs to be investigated. The main disadvantage of all CNT e-emitters is the relatively low output beam currents, which do not exceed a few milliamps. However, large e-beam areas (up to two-inch and more) and their operation in cw- mode indicate the possibility of the wide application of these emitters.

### 4.3. Electron Guns with Plasma Cahodes

The maximum e-beam currents—up to several amperes and above—can be achieved using so-called e-guns with “plasma cathodes”, based on the extraction of the e-beam from the plasma generated either in a hollow cathode discharge [72,73,74,75] or in a surface discharge near the ferroelectric surface [76,77,78,79,80,81,82,83,84,85,86]. Both e-guns with hollow and ferroelectric cathodes do not have thermionic cathodes or other heated parts.

#### 4.3.1. e-Source with a Hollow Cathode

The e-guns based on a high-voltage discharge with a hollow cathode can be operated in both cw- and pulsed discharge modes. Figure 10a shows the electrode configuration of the former consisting of a hollow stainless-steel cathode that is 50 mm in diameter and 100 mm long with a central hole 16 mm in diameter, a plane anode, and a meshed (0.5 × 0.5 mm^2^) tantalum electron extractor [72,73]. To produce an axial magnetic field up to 0.1 T, solenoid coils were used. The working air pressure varied in the forepump range from 0.01 to 0.1 Torr at a residual pressure of 1 mTorr in a vacuum vessel with an e-gun. The current–voltage characteristics of this discharge and extracted e-beam demonstrate the complex dependences on the main parameters of this e-gun. Figure 10b shows that a larger cw-e-beam current up to 400 mA can be reached at a higher discharge current, providing higher plasma density. It is important that this linear dependence was observed at the optimum operating air pressure of 10 mTorr and a magnetic field of 17 mT.

Much higher peak e-currents were achieved in [74], which describes the so-called pseudo-spark pulsed discharge in an e-gun with a plasma cathode, shown in Figure 11a. This e-gun is very similar in appearance to the cw-operated e-gun described above. The discharge occurred at an Ar operating pressure of about 80 Pa in a cylindrical discharge chamber with a height and a diameter of about 59 mm. The measurements revealed the almost linear dependence of the output pulsed e-current on the discharge voltage, and its maximum peak value up to 150 A was observed at a discharge voltage of 20 kV and zero distance from the anode. This value decreased to approximately 100 A at 100 mm from the anode. Importantly, this e-gun can operate without the application of an external guiding magnetic field.

The main advantage of all e-guns with a plasma cathode is their ability to operate at a forepump gas (air or argon) pressure range of ~10^−2^–10^−1^ Torr. These e-guns, without any high-temperature parts, have a long service life and can generate stable e-beams with a maximum cw- and peak e-currents of 400 mA and 150 A, respectively [73,74]. These unique properties of the e-guns with plasma cathodes ensure their promising applications in various fields of plasma technologies, electron-beam melting, laser pumping, etc. However, the need for high-speed vacuum pumping limits the use of electron guns of this type to only powerful stationary installations. We have not been able to find papers describing the use of similar e-guns with hollow plasma cathodes for pumping UVC emitters, but, in our opinion, this may be a very promising application of these e-guns.

#### 4.3.2. e-Guns with a Ferroelectric Cathode

The spontaneous polarization of different ferroelectric materials—LiNbO_3_ [76], lead-germanate [77], lead-zirconium-titanate (PZT or LZT) [78,79], lead-lanthanium-zirconium-titanate (PLZT) [80,81], barium strontium titanate [82], etc.—observed below their Curie temperatures results in screening charges at their surface to keep the charge neutralization at the material boundaries. The spontaneous polarization in these materials can be as high as 20 µC·cm^−2^, resulting in the accumulation of a screening charge density of 10^14^ elementary charges·cm^−2^. Altering the spontaneous polarization rapidly results in the emission of the unbound surface screening charges. This emission mechanism is unique and can be implemented using a patterned metal electrode (strip, grid or ring) deposited on the polar ferroelectric surface, while a solid metal contact is deposited on the rear side, as shown in Figure 12. This type of electrodes induces not only the normal component, but also a tangential one of the applied electric field, which may cause a dramatic effect of accelerating the emitting electrons along the ferroelectric surface, followed by flashover-plasma generation.

The emitted electrons are then accelerated by an e-gun held at a direct current (dc) potential by an external circuit. There are two circuits needed for the experiment: one to provide the gate pulse to induce the electron emission, and the second to provide the electron gun voltage (between the cathode and ground anode) to accelerate the emitted e-beam up to a typical energy of 5–15 keV. These cathodes make it possible to achieve e-beam currents from hundreds of milliamperes to several amperes (with a record electron emission density of up to 1 kA·cm^−2^) for e-beams with an energy of ~10 keV. These e-guns operate in modest vacuum conditions 10^−4^–10^−5^ Torr and, in principle, can be implemented as compact e-beam tubes without continuous vacuum pumping. The lack of reproducibility of the shape of the e-beam current pulse makes it difficult to study such a discharge, and the low service life of the ferroelectric surface is a disadvantage for its practical application.

A theory of the e-emission process from ferroelectrics surfaces has been developed by Schachter et al. [83] and detailed reviews of the physical principles of ferroelectric e-guns have been given by Riege [84], Rosenman et al. [85], Mesyats [86], etc.

## 5. Main Results of UVC Emission Using Electron Beam Pumping

### 5.1. Low-Power UVC Emitters (<20 mW)

#### 5.1.1. Pumping by Focused e-Beams from e-Guns with Thermionic Cathodes

Electron-beam-pumped UVC emitters were first demonstrated in 2009 by Watanabe et al. [87], who studied the CL spectra of hexagonal boron nitride (h-BN) powder. Figure 13 shows that these spectra exhibit broad peaks at 225 nm at room temperature (RT) with an almost linear increase in the intensity to 0.2 mW as the e-beam energy increases to 7 keV. This UVC emitter, pumped by an e-beam with an energy of 7 keV and a current of 50 μA, had a WPE of only 0.6%. Moreover, powder materials have high absorption values and a low probability of radiative recombination upon e-beam excitation. Therefore, in subsequent studies, the main efforts of researchers were focused on the development of UVC emitters based on epitaxial layers and heterostructures in an (Al,Ga)N material system.

In 2010, the Kawakami group published the first results on e-beam-pumped UVC emission from an Al_0.69_Ga_0.31_N∕AlN MQW structure grown on a *c*-sapphire substrate with an AlN buffer [88]. Using an e-beam generated by a thermionic e-gun with a maximum e-energy of 10 keV and a e-current of 50 µA, these authors demonstrated UVC radiation at 238 nm and an output optical power of 100 mW, as shown in Figure 14. However, some doubts should be noted about the correctness of the measurement of optical power in this work, since it claims an anomalously high energy conversion efficiency of 40%, while other authors in numerous later works report significantly lower values (<1% for this type of e-gun). This discrepancy can apparently be due to an erroneous measurement in [88] due to the placement of the UVC radiation detector in a vacuum volume, where its readings could be affected by secondary electrons and X-ray irradiation [60].

This conclusion is confirmed by Shimahara et al. [89], who in 2011 demonstrated compact UVC emitters based on Si-doped (4 × 10^17^ cm^−3^) AlGaN layers which were 800 nm thick, grown by MOCVD on AlN/*c*-Al_2_O_3_ templates. The best layer emitted UVC radiation for 2000 h at 247 nm with an output power of 2.2 mW at an e-beam pumping current of 100 μA and an accelerating voltage of 10kV, which corresponded to an energy conversion efficiency of 0.24%. Later, in 2013, this group demonstrated a UVC emitter based on an Al_0.60_Ga_0.40_N/Al_0.75_Ga_0.25_N MQW heterostructure with a total thickness of 600 nm, which emitted at 256 nm with an output power of up to 15 mW and a WPE of 0.75% [90].

All the initial results on the development of UVC emitters described above were obtained using thermionic e-guns, and the low efficiency of these emitters (<1%) was attributed to the high power of the thermal filament. A greater efficiency (up to 4%) was found by Matsumoto et al. [66] in 2012 for a UVC emitter pumped by a field emission (cold) cathode with graphene nanotips, described in Section 4.2. Figure 15 shows the CL spectra of the 10 × {Al_0.7_Ga_0.3_N(3 nm)/AlN(3 nm)} MQW heterostructure, peaking at 240 nm with a maximum power of 20 mW supplied by this cold cathode e-gun with an e-beam energy and a current of 7.5 keV and 80 µA, respectively. All the spectra measured at various e-energies exhibit not only main peaks near 240 nm, but also broad long-wavelength tails around 350 nm with intensities at the maximum about an order of magnitude lower than the intensities of the main peaks. Moreover, as the e-energy increases, a weak peak near 210 nm appears in the CL spectra, which is attributed to the emission of the AlN buffer layer. Figure 15c shows lines for the calculated and experimental dependences of the emission intensity of 240 nm on e-energy. The theoretical curve was plotted using Monte Carlo method to calculate the penetration depths of the e-beams at various electron energies. Unexpectedly, the maximum intensity is reached at energies of 8–10 keV, when the main part of the e-beam energy should be adsorbed in the thick AlN layer, and not in the relatively thin, 90 nm thick surface region with the MQWs. The authors explained this by the diffusion of excitons formed in the AlN layer at a distance of about 1 μm to the AlGaN/AlN MQW, but, in our opinion, this explanation needs additional studies.

The results of this work indicate the need to match the number of QWs and the total thickness of the active part of the heterostructure with the depth of the e-beam penetration into the structure, which depends on the e-energy, as shown in Figure 3. In this case, it is necessary to take into account the efficiency of carrier transport from barrier layers to quantum wells, which is characterized by the diffusion length of nonequilibrium charge carriers in layers in the (Al,Ga)N system. The known literature data refer to measurements of this parameter, most often in binary GaN layers, using various experimental techniques. Their values vary in the range from 30 nm to 3 μm depending on the structural quality of the layers, which is largely determined by the epitaxy technology used [91,92,93,94,95,96]. For MOCVD-grown layers on *c*-sapphire substrates, most of the diffusion lengths lie in the range of 100–200 nm. The GaN layers grown on the same substrates using PA MBE have smaller diffusion lengths. For example, in a recent paper by Brandt et al. [95], the diffusion length of carriers for a similar layer grown using PA MBE on a MOCVD-grown GaN (0001) template was experimentally determined to be 40 ± 5 nm, and in the work of Daudin’s group, this parameter for an Al_0.27_Ga_0.73_N layer, also grown using PA MPE on an AlN/6H-SiC template, was estimated at 7.5 nm [97].

The next issue discussed in [66] is the dependence of internal quantum yield on the excitation level, which turned out to be proportional to the e-beam current density. Therefore, based on this dependence, it is desirable to increase this parameter. However, at high values of this parameter, efficient heat removal from the local region of the heterostructure irradiated by an e-beam is problematic. The use of e-beams of a small diameter, scanning a sufficiently large area of the structure, makes it possible to solve both the problem of heat removal and the problem of a small internal output simultaneously.

#### 5.1.2. UVC Emitters Pumped by CNT-Based e-Sources 

The development of large-area e-sources with CNT-based cathodes, described in Section 4.2, allowed the Park group to demonstrate in 2019 the use of this e-gun for the pumping of UVC emitters based on AlGaN heterostructures [68]. Figure 16 shows the main characteristics of a UVC emitter pumped by such a CNT e-source with an e-beam current of 1.0 mA over an e-irradiation area of 188 mm^2^ at an anode voltage of 3 kV. When pumped by this e-source, the MOCVD-grown Al_0.47_Ga_0.53_N/Al_0.56_Ga_0.44_N structure on a *c*-sapphire substrate exhibited UVC radiation at 278.7 nm with a maximum output optical power density of 0.11 mW·cm^−2^ (~0.3 mW) on a light-emitting area of 303 mm^2^.

In the next work of this group [70], an e-source with an arranged array of nano-tipped CNT cathodes and an e-emitting area of 276 mm^2^ was used to excite an Al_0.73_Ga_0.27_N(2 nm)/AlN(8 nm) MQW heterostructure. Figure 17a shows its CL spectra with the main peak at 233 nm, which can be attributed to near-band-edge (NBE) emission. The intensity of this peak increased rapidly as the e-energy increased from 4 to 5 keV, as shown in Figure 17b. The minimum energy of an e-beam for observing MQW-related CL was explained by the presence of a 170 nm thick Al film on the structure surface, which the electrons must have sufficient energy (>3 keV) to overcome. Moreover, as the electron-beam energy increased, the intensity of the broad peak at about 330 nm increased. This peak was attributed to defect states associated with V_Al_ complexes (V_Al_-oxygen) of the AlN layer under the upper MQWs [98].

Figure 17b,c shows the dependences of CL intensities on the e-beam parameters, revealing the saturation effect with a maximum power density of 2.3 mW·cm^−2^, both with an increase in the e-energy above 5 keV and with an increase in the e-beam current above 0.5 mA. In addition, a weak “red” shift of 37 meV (2 nm) for the CL peak spectral position and its widening (not shown) were found with an increase in e-energy. This tendency of AlGaN-based MQWs is opposite to that of InGaN/GaN MQWs and can be explained by the thermal effect and composition fluctuations [99,100,101]. Namely, the observed effects mean the redistribution of thermally excited electrons into deep localized states, which leads to a red shift of the CL peak and an increase in its width.

### 5.2. Medium-Power UVC Emitters (Up to 1 W)

#### 5.2.1. UVC Emitters Pumped by Thermionic e-Guns

All AlGaN-based MQW structures for e-beam-pumped UVC emitters described in Section 5.1 were grown using MOCVD. Similar structures grown using PA MBE were first demonstrated in 2015 by Ivanov et al. [102]. A distinctive feature of this work was the method of growing Al_x_Ga_1−x_N QWs in the form of *N* × {(GaN)_m_/(Al_y_Ga_1−y_N)_n_} heterostructures with the introduction of several (*N*) ultra-thin GaN inserts with a nominal thickness of *m* monolayers (1 ML~0.25 nm) into Al_y_Ga_1−y_N barrier layers with a nominal distance between GaN inserts of *n* MLs, as initially suggested in [103]. The 40 × {Al_0.6_Ga_0.4_N(2.2 nm)/Al_0.7_Ga_0.3_N(38 nm)} MQW structure with the wells grown in the form 3 × {(GaN)_0.5_/(Al_0.7_Ga_0.3_N)_3.5_} exhibited UVC radiation at 270 nm and maximum peak (cw-) output optical powers of up to 60(4.7) mW at an WPE of 0.24%, while the internal quantum efficiency of the MQW structure was 50%. The measurement scheme is shown in Figure 2c.

Later, in 2016, Wang’s group used this method [104] to fabricate UVC emitters, 40 × {3 × [(GaN)_1_/(Al_0.75_Ga_0.25_N)_2_]/Al_0.75_Ga_0.25_N(31 nm)}, emitting at 285 nm with peak (cw-) output optical powers of 110(25) mW and an maximum WPE of 0.6%, whose dependences on the parameters of e-beams are shown in Figure 18.

In 2016, Tabataba-Vakili et al. [60] used MOCVD to fabricate 10 × Al_0.56_Ga_0.44_N/Al_0.9_Ga_0.1_N MQW structures with a well thickness of 1.4 nm. These structures, when pumped by e-beam with an energy of 12 keV, emitted at 246 nm with peak (cw-) maximum output optical powers of 230(30) mW at a maximum e-beam current of 4.4 mA, as shown in Figure 19a,b. The WPE for the test sample was 0.43% (at 12 keV), and examination of the several factors contributing to the overall efficiency yielded an estimate for the internal quantum efficiency of about 23%. Moreover, Figure 19c shows the characteristics of the CL spectra excited by an e-beam of different diameters of 3 mm or 50 µm. The lower CL intensity in the case of the focused e-beam was explained by a decrease in the internal quantum yield due to greater heating of the active region by an e-beam of a smaller diameter. However, this explanation does not seem to be convincing since the linear dependences of the output optical powers on the e-beam currents were observed for both e-beam diameters.

After the first e-beam-pumped UVC emitters with Al_x_Ga_1−x_N/Al_y_Ga_1−y_N MQW structures with a standard QW thickness of 1.4–3 nm, the next step in the development of these devices was a transition to the use of light-emitting *N* × {GaN_m_/AlN_n_} MQWs heterostructures with a monolayer (1 ML~0.25 nm) and even fractional thicknesses of the wells. The possibility of efficient light emission by such structures in the UV range was theoretically substantiated in 2011 by Kamiya et al. [105], and almost immediately implemented by Taniyasu and Kasu using MOCVD technology [106]. Later, in 2016, such MQW structures were grown by Jena’s group using PA MBE technology [107]. They used either standard or modified Stranski–Krastanov growth modes under N-rich conditions (Ga/N_2_* < 1) to induce the formation of 3D GaN islands (quantum dots/disks) into GaN QWs with nominal thicknesses of 1–3 MLs [108]. Despite the enhanced PL intensity, high IQE and TE-polarization of the output UVC emission from these heterostructures, the maximum achieved output power density of the related UVC-LEDs at the shortest wavelength of 234 nm was below 0.4 mW·cm^−2^, which was attributed to the insufficient *p*-type doping of these devices [109]. 

The first ML-thick GaN/AlN MQW structures for e-beam pumping with a number of periods from 40 to 360 (with a total structure thickness of ~1.8 μm) were grown using PA MBE in 2018 [110]. The main distinguishing feature of their fabrication was the use of metal-enriched conditions for the growth both of barrier layers and of quantum wells with typical flux ratios of Al/N_2_*~1.1 and Ga/N_2_* up to 2.2 at relatively low substrate temperatures of 690–700 °C, as shown in Figure 20a.

During the growth of structures, excess Al was consumed due to short-term exposures of the surface under plasma-activated nitrogen flux before the growth of each QW, and excess Ga was evaporated during the subsequent growth of barrier layers, since the growth temperature provided a rather high Ga evaporation rate of 0.2–0.3 ML s^−1^ [111].

The used Me-rich conditions at a low growth temperature make it possible to achieve very sharp interfaces in ML-thick QWs due to the kinetic limitation of the Ga segregation effect in the heterostructure, which plays a significant role at higher temperatures. Second, these conditions ensured the precise control of the nominal thicknesses of both QWs and barrier layers using calibrated data on N_2_*-flux [111]. The sharpness of the symmetrical interfaces in the MQW structures and accuracy of setting their parameters were confirmed using images from high-angle annular dark field scanning transmission electron microscope (HAADF STEM) and high-resolution X-ray diffraction (HRXRD) ω-2θ scans of (0002) symmetric reflection [110].

Figure 20b demonstrates narrow CL peaks with a spectral width of ~20 nm in the UVC range of 235–250 nm for *N* × {GaN_1.5_/AlN_m_} MQW structures with *N* = 40−360 and *m* = 22−154, excited by a thermionic e-gun with an electron energy of 20 keV. From them, the linear dependences of output peak optical power on the pump current follow, and its maximum values of 150 mW are achieved at a pulse-scanning mode with an e-current of 1 mA of the structure with the largest number of QWs, as shown in Figure 20c. At the same time, the maximum WPE value of 0.75% is also observed in this structure. The maximum cw- output power in this structure reached 28 mW when it was excited by an e-beam with *E*_e_ = 15 keV and a diameter of 10 mm at an e-current of 0.45 mA (i.e., WPE was 0.4% in this excitation mode).

In 2019, the Monroy group launched a series of works on the development of e-beam-pumped UVC and UVB emitters based on AlGaN/AlN MQW structures grown using PA MBE [112,113,114]. In the first works [112,113], nanowire structures were grown starting with the growth of base GaN nanowires with a height of 900 nm, followed by 88 × {Al_x_Ga_1−x_N(*x* = 0, 0.05, 0.1)/AlN} MQWs having a well thickness of 0.65–1.5 nm, while the barrier thickness exceeded 3 nm. All parts of the nanowires were grown under the same N-rich conditions with a flux ratio Ga/N_2_* of 0.25 at a substrate temperature of 810 °C, and a growth rate of 330 nm·h^−1^. Figure 21a,b shows transverse- and plan-view images of these structures obtained using a scanning electron microscope (SEM), which indicates the formation of separate nanocolumns with a surface density 6–8 × 10^9^ cm^−2^ and a diameter of 30–50 nm.

Figure 21c,d shows HAADF STEM images of individual QWs confirmed the given values of layer thicknesses and periods of the MQW structures. The latter parameter was also confirmed by XRD analysis (not shown). Finally, Figure 21e,f shows the CL-RT spectra of these structures, with single peaks showing a shift from 260 to 340 nm as the QW thickness increases from 0.65 to 1.5 nm (from 2.5 to 6 ML), respectively, while maintaining the peak shape. The ratio of the integrated luminescence intensity at RT and at low temperature was used to estimate the internal quantum efficiency (IQE) of these structures. The measurements of PL and CL spectra revealed an IQE = 63% for the structure emitting at 340 nm, and it decreased to ≅ 22% for the structure emitting at 258 nm. In the next work, Harikumar et. al. [114] studied planar superlattice (SL) heterostructures with Al_x_Ga_1−x_N/AlN (*x* = 0 or 0.1) quantum dots (QDs) with a density of >10^11^ cm^−2^, grown using PA MBE with a total thickness of 530 nm (100 periods) on 1 μm thick (0001)-oriented AlN on sapphire templates at a substrate temperature of 720 °C. The growth of GaN and Al_0.1_Ga_0.9_N QDs was carried out under nitrogen-enriched conditions with different flux ratios of Ga/N = 0.29–0.85, while for the growth of barrier layers with a thickness of 4 nm, slightly metal-enriched conditions were used with a flux ratio Al/N = 1.1.

All the structures exhibited AFM images with a relatively smooth surface topography of the structures, as shown in Figure 22a. However, HAADF images of the QWs in these SLs demonstrated an inhomogeneous distribution of their thicknesses with the formation of local, small extensions in the form of QDs, as shown in Figure 22b–d. This inhomogeneous QW/QD topography can be associated with the nitrogen-enriched growth conditions used during QD growth, when their nominal thickness was proportional to the Ga flux. Therefore, as observed in Figure 21e, the short-wavelength shift of the CL-RT peaks from 335 to 244 nm with a decrease in the Ga flux can be caused by a decrease in the nominal thickness of the QDs and a corresponding decrease in their average size. A blue shift is observed in the peak emission wavelengths of Al_0.1_Ga_0.9_N dots as compared to GaN dots with the same Ga flux. The shift corresponds to an average increase of 250 meV in the band gap, which is consistent with the incorporation of 10% of Al in the dots.

It is important that the internal quantum efficiency of UVC emitters with GaN (or Al_0.1_Ga_0.9_N)/AlN QDs, which was estimated in the first approximation from the ratio of the PL intensities at room and zero (extrapolated) temperatures, was 50% for the structures grown at the highly nitrogen-enriched conditions with a flux ratio Ga/N_2_* of <0.75. It should also be noted that such a high efficiency was observed in a wide range of power densities of PL laser excitation up to 200 kW·cm^−2^. In addition, the same spectral position of the main CL peak was shown with an increase in the pumping electron energy, as shown in Figure 23a. However, it should be noted that at energies above 20 keV, an additional long-wavelength peak appeared at 340 nm, which was associated with the penetration of a high-energy e-beam into the AlN buffer layer, in which it excites defective radiation (associated with carbon atoms and other defects).

Figure 23b demonstrates an almost linear increase in the CL intensity as the e-energy increases to 10 keV, which was explained by a simple increase in the active (light-emitting) region in the structures at a constant excitation efficiency, as shown in Figure 23c. The CL intensity began to fall at electron energies above 10 keV, which was accompanied by a drop in the efficiency. This effect was explained by the penetration of the exciting beam into the substrate and the charging of dielectric AlN. Finally, a linear increase in the CL intensity was observed up to the maximum possible e-beam current up to 0.8 mA for both low and medium e-energies. However, at a higher electron energy (~10 keV), this dependence demonstrated the saturation effect at high e-currents, which was also explained by surface charging. Unfortunately, papers [112,113,114] did not report on the absolute values of the output optical powers from the studied heterostructures.

Recently (in 2023), two research groups using PA MBE technology have joined the study of ML-thick GaN/AlN MQW structures emitting in the UVC range with e-beam pumping [115,116]. The first of them, headed by Daudin, is the leading scientific group that has developed a basic understanding of the PA MBE growth kinetics of III-N layers and associated heterostructures [117,118]. In particular, since the late 1990s, they have been studying the features of the growth of GaN/AlN heterostructures, but so far, their main attention has been focused on the formation of quantum dots in this system according to the Stranski–Krastanov mechanism, the radiation of which is outside the UVC range. This group was the first to measure the critical thickness of 2 ML for the 2D–3D transition during the growth of GaN over AlN [119]. In addition, they carried out pioneering studies of segregation phenomena during the growth of GaN/AlN heterostructures, which determine the temperature dependences of the sharpness and symmetry of heterointerfaces in these structures [120]. In a recent work of this group [115], the fine structure of the CL spectra of 1–4 ML thick GaN disks in AlN nanowires was studied to control the emission wavelength of AlN nanowires. Particular attention in this work was paid to emission lines below 240 nm (which correspond to 1 ML), which were assigned to the recombination of confined carriers in incomplete QWs with a lateral size smaller or comparable to the 2.8 nm GaN Bohr exciton radius. Their emission consists of sharp lines extending up to 215 nm near the edge of the AlN band, as shown in Figure 24.

The RT-CL intensity of an ensemble of GaN quantum disks embedded in AlN nanowires is about 20% that of the low temperature value, emphasizing the potential of ultrathin/incomplete GaN quantum disks for deep UV emission. The GaN/AlN MQWs were grown using alternating exposure to metal and nitrogen fluxes at the relatively high growth temperature of 750–800 °C, allowing eased metal diffusion on the top surface prior to nitridation of the metallic layer.

In contrast, Araki’s group grew a GaN/AlN superlattice with a well thickness estimated at 1.1–1.4 ML, using PA MBE with metal-rich growth conditions for both group III atoms, which provided an atomically flat AlN surface and abrupt interfaces in these heterostructures [116]. Subsequently, excess metals were eliminated by the so-called method of droplet elimination by radical beam irradiation for AlN (originally, this method was developed by this group for InN [121]) and growth interruptions for GaN. These superlattices exhibited CL spectra with a peak wavelength of 230–260 nm at RT, as expected. The emission wavelength shifted with an increasing thickness of the AlN layer.

#### 5.2.2. Sub-Watt Power UV Emitters Cw-Pumped by Large-Area CNT-Based e-Sources 

The possibility of exciting UV-emission from a sapphire plate was initially demonstrated in 2020 by Park’s group using large-area e-sources based on the cold field CNT cathode shown in Figure 2e [58]. Fully vertical, aligned cone-shaped CNT emitters are arranged periodically at 30 μm intervals to enhance the electron emission current. The one island consists of 49 CNT emitters in a square shape. These islands are patterned at 0.5 mm intervals to reduce cathode current leakage through the gate electrode.

Figure 25a shows that the CL spectra of this plate exhibited several lines in a wide spectral range above 200–400 nm at low electron energies (5–9 keV), but at high electron energies (10 keV), only one main broad peak at 226 nm is observed. The output optical power, integrated over the spectrum and emitting area of 960 mm^2^, was 113 mW at a WPE of 0.87%. However, the very wide emission spectrum of such a UV emitter up to the visible range limits its application.

In another paper published in 2021, Mohan et al. [71] used a large-area CNT e-emitter, shown in Figure 9, in the development of UVA emitters based on GaN/AlGaN MQW structures. Figure 26 shows the CL spectra and the dependence of the output power on the pumping current for a UVA emitter based on a 18 × {GaN(2.1 nm)/AlGaN(9 nm)} MQW structure emitting at 330 nm. The defect luminescence (<10%) in the visible region constitutes the well-defined quality of the AlGaN/GaN MQWs grown on an AlN/sapphire substrate. When it was pumped in a cw- mode by an e-beam with energy of 7 keV and a maximum current of 1 mA, a maximum output optical power of 225 mW was achieved with an as-calculated quantum efficiency of 3.6%.

CNT-based DC-operated triode UV emitters do not require a pulsed operation mode and can be easily operated with minimal power adjustment. These devices are compact, simple and easy to use and manage, requiring minimal maintenance and greater efficiency.

The successful development of a UVC emitter pumped by a large-area (two-inch) CNT e-source and increased WPE was recently demonstrated by Wang et al. from Peking University [122]. They proposed a specially designed new type of ML-thick GaN/AlGaN/AlN MQW structures. The main distinguishing feature of these emitters is a new QW design, in which a 2 ML thick Al_x_Ga_1−x_N layer is introduced before one ML thick GaN QWs according to the formula 100 × {GaN_1_/(Al_x_Ga_1−x_N)_2_/AlN_40_} (*x* = 0.6, 0.5, 0), as shown in the inset in Figure 27a.

The authors calculated that the maximum overlap degree of the wavefunctions of electrons and holes in these complex QWs is reached at *x* = 0.6 and amounts to 0.85. Figure 27a confirms this theoretical result by showing the CL spectra of three different MOCVD-grown MQW structures and revealing the highest CL intensity for the proposed optimal design. Importantly, this structure emits at 248 nm with an output optical power controlled by both anode voltage and e-current, as shown in Figure 27b,c.

A maximum power value of 702 mW was achieved at an anode voltage of 7 keV and an anode current of 3 mA. However, Figure 27d shows that the maximum value of the WPE, equal to 5.2%, was observed at a lower current of 1 mA and the same anode voltage, corresponding to an output power of about 300 mW. Note that this WPE value is more than five times higher than the best values of this parameter for sub 250 nm UVC-LEDs.

### 5.3. High-Power (Up to Several Tens of Watt) Pulsed UVC Emitters

A watt level for output peak optical power for electron-beam-pumped UVC emitters was overcome in 2019 by Wang et al. [123]. This group was the first to transfer the idea of ML thick *N* × {GaN_m_/AlN_n_} MQWs structures (*m* = 1–3) from PA MBE to MOCVD technology and used this to pump these heterostructures with relatively high current (up to 40 mA) pulsed thermionic e-guns with LaB_6_ cathodes. Figure 28a shows high-resolution X-ray diffraction (HRXRD) 2θ-ω scans along the symmetric (0002) plane, which revealed more than 14 satellite peaks, indicating satisfactory periodicity and sharp interfaces in the MQW structures. Their simulation confirmed a barrier thickness of about ≈10.2 nm, and a well width of ≈0.3, 0.45 and 0.72 nm, being in good agreement with the designed values of 1, 2 and 3 ML, respectively. An atomic force microscopy (AFM) study of the surface topography of these structures showed distinct atomic steps on the surface, with a typical RMS surface roughness (RMS) of 0.32 nm in a 3 × 3 µm^2^ scan area, as shown in Figure 28b. Moreover, Figure 28c shows HAADF measurements of MQW structures with different nominal QW thicknesses, which are recognized by the bright stripes, although a slight interdiffusion can be seen in the TEM images between the well and top barrier layers. This is mainly caused by the relatively high growth temperature of MOCVD. The well thicknesses for the three samples are one to two MLs, two to three MLs, and three to four MLs, which is in good accord with the XRD results.

The MOCVD-grown structures with nominal QW thicknesses varying from 1 to 3 ML exhibited CL in the range of 230 to 270 nm, as shown in Figure 28d. It is important that the intensity of the short-wavelength CL was more than an order of magnitude lower than the intensity of the long-wavelength CL. After optimization of the design of the UVC emitters, a 150 × {GaN_2_/AlN_40_} MQW structure was grown, which demonstrated a CL single peak at 258 nm with a maximum pulsed optical power of up to 2 W at an e-beam supplied by an e-gun with a LaB_6_ cathode, which provided an e-beam current of 37 mA and energy of 18 keV, as shown in Figure 28e,f.

At the same time, the development of PA MBE to obtain ML thick MQWs continued, and the results of growing a large series of such structures with different QW thicknesses and growth conditions were published in the period 2021–2023 in [124,125]. Features of the growth of PA MBE AlN/*c*-Al_2_O_3_ templates for these structures, including the issues of reducing the density of threading dislocations in these templates to ~5 × 10^−9^ cm^−2^ and the development of elastic stresses in them, are described in [126,127]. Figure 29a shows the first CL spectra of these samples with a QW thickness of *m* = 1.25–7 MC [124].

Topography studies of the 350 × {GaN_1.5_/AlN_16_} structures grown in Me-rich conditions revealed signs of spiral growth with the formation of a terrace-like surface with steps of 1–2 ML in height and a width (inter-step spacing) of ~20 nm, as shown in Figure 30a,b. Despite the formation of a surface with a nano-hillock topography, its root-mean-square roughness was 0.3–0.7 nm and its values weakly depended on the AFM scanning area, which varied from 1 × 1 µm^2^ to 5 × 5 µm^2^. The main factor influencing the surface roughness was the value of the Ga/N_2_* flux ratio used during QW growth. During QW growth under the weakly metal-enriched conditions with a Ga/N_2_* of ~1.1, the surface corresponded to a mixed growth mechanism with signs of both a step-flow and 2D nucleation growth mechanisms, which led to the formation of a surface with a characteristic step height of 1 ML and minimum values of rms of 0.3 nm (see Figure 30a). In the case of a high value Ga/N_2_* ~2), the surface to a greater extent corresponded to the step-flow growth mechanism with the formation of steps with a height of 2 ML, as shown in the inset of Figure 30b. Moreover, the growth of QWs under the N-rich conditions corresponded to a Ga/N_2_* = 0.6 and the surface exhibited the roughest 3D surface topography without any evidence of a stepped topography (see Figure 4c,d in [125]).

Figure 30c shows a HAADF STEM image of the internal morphology of one of the MQW structures grown at Ga/N_2_* = 2.2, which confirms, first, the nominal period of this structure [125]. In addition, this image corresponds to the AFM image of the surface and indicates the formation of stepped cross-section profiles of the QWs with regions of different thicknesses, both 1 and 2 ML, which corresponds to their fractional (1.5 ML) nominal thickness.

Studies of the structures using HRXRD with scanning of the symmetric reflection (0002) also confirmed the nominal values of the MQW parameters of the structures [124,125]. However, the measurements of longitudinal and transverse scans of X-ray reflection curves (XRR), which are shown in Figure 30d,e, respectively, turned out to be much more informative for the characterization of QWs. These scans confirmed the smoothest morphology of both the surface and QWs in structures grown under weak metal-enriched conditions (Ga/N_2_* = 1.1) according to a mixed growth mechanism with minimal AFM roughness and a characteristic step height of 1 ML. The higher surface and QW roughness found for the structures grown at Ga/N_2_* = 2.2 was also consistent with the AFM measurement data. And, finally, the roughest morphology was found in structures with QWs grown under nitrogen-enriched conditions (Ga/N_2_* = 0.6) that correspond to the AFM data described above.

The CL spectra in these heterostructures were excited using various high-current e-guns. The initial studies were carried out using a thermionic e-gun with a LaB_6_-based cathode emitting an e-beam with a diameter of ~1 mm and a maximum pulsed e-current of up to 60 mA current with a pulse duration of about 50 ns [124]. In addition, in both works, an e-gun with a cold plasma cathode based on a surface discharge on a ferroelectric surface was used, which supplied the emission of e-beams of 4 mm in diameter with maximum currents up to 500 mA [124] and 2.0 A [125]. Figure 31a shows two CL spectra of the 360 × {GaN_1.5_/AlN_22_} structure measured using these types of e-guns for their excitation. Both spectra exhibit single peaks at 241–242 nm with half-widths of 12–13 nm. The peak intensity of the output optical power of the CL excited by the LaB_6_-based e-gun depended linearly on the e-beam current, demonstrating a maximum output optical power of 1 W at a pump current of 65 mA [124], which approximately corresponds to the data obtained in [123] for a similar MOCVD-grown structure, described above. In the e-guns with a plasma ferroelectric cathode, the temporal shape of the e-beam pulse has a complex character, as shown in Figure 31b, where the lower curve shows a sequence of several short e-beam current pulses with a maximum current up to 1.8 A and a total duration of 0.5 µs. The upper curve in this figure shows the corresponding change in the output optical power. By integrating the output power pulse, the average energy of each pulse can be calculated. Figure 31c shows the linear dependence of this parameter on the discharge voltage (e-energy) in an e-gun with a plasma cathode, the maximum value of which is 5 μJ per pulse at a voltage (e-energy) of 12.5 kV (keV).

Studies of the optical properties of 400 × {GaN_1.5_/AlN_16_} MQW structures grown at different Ga/N_2_* ratios and templates obtained either by MOCVD or by a combination of MOCVD and PA BE were carried out using measurements of the PL and CL spectra excited by 4th harmonic of a Ti-sapphire laser and various types of e-guns emitting e-beam currents in the range from 30 nA up to 2A, respectively [125]. All the spectra showed a maximum blue shift from 260–270 nm to 230–240 nm, as well as the narrowing of single peaks for the MQW structures grown at a lower Ga/N_2_* ratio. Comparison of these spectra with the spectra of test GaN_m_/AlN_16_ MQW structures with integer values of QW thicknesses *m* = 1 and 2, as well as with data from studies of the structural properties of MQW structures by AFM, XRR and HAADF STEM, led to the conclusion that various quantum-size objects are formed depending on the growth conditions. In the case of large values of Ga/N_2_*, the formation of local islands occurs with an effective thickness of 2 ML on atomically smooth terraces with a lateral size of 20–30 nm up to the whole width of the terrace. In contrast, the structures grown at a reduced Ga/N_2_* ratio of 1.1 exhibited the formation of higher-density quantum disks with a thickness of 1 ML and smaller lateral dimensions, which corresponds to a transition from a pure spiral step-flow growth mode to a mixed-growth mode with a reduced adatom mobility.

The optical properties of the ML-thick QW structures are determined by excitons up to RT, as was predicted theoretically [107] and experimentally demonstrated in [130]. It should also be noted that there is a relatively small difference between the PL and CL spectra for the MQW structures grown on AlN templates formed either by PA MBE only or sequential MOCVD and PA MBE. This implies that the characteristic size of charge carrier localization regions is much smaller than the average distance between the defect centers, and for quantum disks, the distance to a spiral is of no importance.

Finally, the analysis of the CL spectra excited by a plasma cathode e-gun allowed one to plot the dependences of peak output optical power on the e-beam current, which are presented in Figure 31e. These power dependences are linear for all the MQW structures within all ranges of output optical power variations, i.e., the droop regime was not yet attained. Figure 31e shows that the highest pulse power up to 50 W at a wavelength of 267 nm was achieved for the structure grown under the highly metal-enriched conditions on a MOCVD/PAMBE AlN template. The same MQW structures grown on the PA MBE AlN template emitted power at almost the same wavelength of 265 nm, and demonstrated somewhat lower power values of about 35 W. The maximum power value of 10 W at a wavelength of 238 nm was observed for structures with 1.5 ML thick MQWs grown under a low Ga/N flux ratio of 1.1 on both types of AlN/*c*-Al_2_O_3_ templates.

The results demonstrated above showed that e-beam pumping of ML thick GaN/AlN MQW structures allows one to obtain a high-intensity UVC emission in the spectral range of 240–260 nm with peak optical powers up to several Watts and more. However, the wall-plug efficiency of the first e-beam-pumped UVC emitters was extremely low, only 0.1–0.3%, and further optimization of the MQW structure design and the e-beam pumping scheme are required to improve this parameter.

A detailed overview of the physics and fabrication technologies of the monolayer-thick GaN/AlN MQWs, from their ab initio calculations to the implementation of various UV-light-emitting devices, can be found in [56].

The progress in the development of e-beam-pumped UVC emitters of various designs and grown using different technologies is presented in Table 2.

## 6. Preliminary Analysis of Stability of UVC Emitters with e-Beam Pumping

One of the most urgent problems of UVC emitters is to achieve their reliable long-term operation at RT under the action of fairly intense e-beams with an energy of 5–20 keV on light-emitting heterostructures in an (Al,Ga)N system. In this case, this problem must be solved both for the e-beam source and for the semiconductor heterostructure.

The stable operation of e-guns is determined primarily by the lifetime of the cathodes, which, as shown in Table 1, ranges from several hundred to a thousand hours for thermionic cathodes. The lifetimes of cold emission cathodes based on carbon materials have been studied much less, but Matsumoto’s research paper notes that the emissivity of graphene nanotips remains unchanged for up to 5000 h and even more [66]. This corresponds to the maximum value of this parameter among all those reported. The mediocre stability of plasma cathodes based on an e-emission from ferroelectrics should be noted, which retain stable surface properties only for <10^7^ pulses of 0.2 µs in duration, and additional studies are needed to increase this parameter [85].

To ensure the long-term operation of any light-emitting heterostructures with various types of pumping, it is necessary to provide a low level of structural defects. This is not an easy task, because of the relatively high level of various bulk and point defects that occur during all stages of growth of AlGaN-based UVC optoelectronic devices. At the initial growth stages, the generation of high elastic stresses in heterostructures, usually grown on highly mismatched sapphire substrates, leads to a high density of threading dislocations (~10^9^ cm^−2^), most of which easily reach the active (light-emitting) regions through a few µm thick AlN buffer layers [2]. Numerous point defects are introduced into the layers at all stages of growth. These defects include various crystallographic defects in the host III-N lattice (primarily, vacancies of type *V*_III_, *V_N_*), different impurity atoms from the residual atmosphere (C, O, etc.), as well as complexes between these defects of various types [131,132]. In addition, in conductive emitters in UVC-LEDs, various complexes of point defects are formed with the participation of crystallographic defects and *p*-type (Mg) or *n*-type (Si) impurity atoms [13,17,133,134]. It is important that the concentration of all the defects described above increases with an increasing Al content in the AlGaN compounds.

All the described-above defects play an important role in degradation processes in UVC-LEDs. The lifetime of these devices emitting at 275 nm does not exceed several thousand hours (<10,000) [4]. The situation is even worse for far-UVC-LEDs. For example, in a recently (2022) published work a research-grade far-UVC-LED emitting at 233 nm showed a relative optical power reduction of up to about 40% over 250 h of operation at a stress current of 20 mA and to <10% for a stress current of 100 mA [135]. Thus, the lifetimes of UVC-LEDs are significantly inferior to visible LEDs, which currently have a lifetime of 30,000–50,000 h [4].

The absence of the need to dope e-beam-pumped UVC emitters, as mentioned above, is an advantage of these devices. However, on the other hand, the use of energetic particles (electrons) for pumping this type of emitters leads to a modification of the structural quality of light-emitting heterostructures. In principle, this should be accompanied by a deterioration in their optical properties, including a decrease in the output optical power and a change in the shape and position of the CL spectra. Unfortunately, there is no specific information about this phenomena in the case of the interaction of relatively low-energy (<20 keV) e-beams with (Al,Ga)N-based MQW heterostructures in the known literature. Therefore, the possible processes of this interaction are discussed below only on the basis of general considerations.

A theoretical analysis using ab initio molecular dynamics calculations of the elastic interaction of electrons with wurtzite GaN and AlN lattices, performed by Weber’s group [136,137], showed that the minimum threshold displacement energy for Ga atoms is 39 eV, while for Al atoms, this energy increases to 55 eV. For nitrogen atoms, the analogous threshold displacement energies are 17 and 19 eV in GaN and AlN, respectively. The lattice atoms, having acquired these energies, are able to leave their regular positions and form various defects. The Weber group describes various configurations of Frenkel defects in binary GaN and AlN layers that appear under an electron beam. The lowest threshold displacement energies have defect configurations consisting of nitrogen vacancies (*V*_N_) and nitrogen–nitrogen (*N*–*N*) dumbbells in GaN or single interstitial defects in AlN. These irradiation-induced defects influence the electronic structures of the compounds. The existence of Frenkel pairs induces localized defect states within the bandgap of the compounds, which can trap electrons or holes, and thus affect the optical and transport properties of the heterostructures. However, in order to transfer these relatively high threshold displacement energies to lattice atoms, the electrons in the primary beam must have a significant energy, and, as shown in [137], its value must significantly exceed 100 keV. Therefore, e-beams with energies above 10 MeV are usually used to significantly modify the electrical properties of AlGaN heterostructures [138]. In addition, this type of defect generation plays an important role in transmission electron microscopy, which mainly uses focused e-beams with energies of several hundred keV [139].

Thus, the contribution of the processes of defect formation through the direct kinetic displacement of host lattice atoms by e-beams with a relatively low energy (<20 keV), which are usually used to pump UVC emitters, is insignificant. Furthermore, it is possible to ignore the possible processes of sputtering of (Al,Ga)N compounds by an e-beam since the threshold energies of this process are even higher [139].

However, the e-beam bombardment of heterostructures, along with elastic interaction electrons with the hosting atoms of the crystal lattice, can also structurally modify defects already present in heterostructures, which will lead to a change in the optical and transport properties of both QWs and barrier layers. In addition, QWs themselves, which usually have high elastic stresses, cluster-like formations, etc., can be considered as “defects” (i.e., areas with a locally distorted crystallographic lattice). In this case, the threshold displacement energies of atoms in defective regions can be significantly lower than the values of analogous parameters for atoms in the hosting lattices. 

Applying this idea, Zverev et al. [140] described a catastrophic degradation of pulsed lasers based on AlGaAs/InGaAs/GaAs heterostructures with e-beam pumping with an e-energy of 26 keV. They suggested the occurrence of irreversible changes in the low-strained active region of the laser at a pumping e-current density of about 100A cm^−2^, while in structures with a higher deformation in the active region, similar catastrophic failures were observed at much lower current densities of <30 A cm^−2^.

In III-nitrides devices, a significant modification of the CL spectra from InGaN/GaN QWs was observed by Shmidt et al. [141], at a relatively low energy of the e-beam of 10 keV and a relatively low irradiation dose of several Cl·cm^−2^. The observed spectra transformation was attributed to the change in the thickness/composition of the QWs due to enhanced diffusion of In atoms. A lower intensity of similar processes in heterostructures in the (Al,Ga)N system due to higher binding energies in these compounds should be noted. Moreover, one can expect weaker electron-enhanced defect formation/modification in the heterostructures based on GaN/AlN binary compounds compared to heterostructures based on ternary compounds, which is indirectly confirmed by studies of high-energy electron action on these two types of heterostructures [138].

Based on the general considerations formulated in [139], in a qualitative analysis of the inelastic electron–electron interaction between beam electrons and an electron system in semiconductors, the following processes should be taken into account.

First, it is necessary to exclude the charging of electrically insulated surfaces such as AlN and Al-rich AlGaN, whose properties are close to those of dielectrics. This process occurs as a result of both partial reflection and secondary electron emission when the surface is bombarded with an electron beam. The most effective way to avoid charge accumulation on the surface is to deposit a metal layer (usually Al) which is several hundred nm thick, which acts as an effective charge sink. Moreover, the metal deposited on the surface of semiconductor heterostructures protects them from contamination by various decomposition products of residual gases (hydrocarbons) arising under e-beam bombardment.

The next important process of inelastic electron–electron interaction is local heating of samples, which usually accelerates the rate of degradation of light-emitting structures. To reduce the thermal load in the case of focused e-beams, a pulse-scanning mode of e-beam movement over a large sample area is usually used. Moreover, pulse-mode excitation is used in the case of a high e-beam current (up to a few amperes [125]).

A search in the literature revealed very few works that studied the stability of the properties of e-beam-pumped UVC emitters. Only Shimahara et al. [89] measured the lifetime of an electron tube based on an AlGaN:Si layer 800 nm thick, emitting at 247 nm. It has been established that the maximum power of 2.5 mW, achieved when using an e- beam with a diameter of 6 mm at an accelerating voltage of 10 kV and a cw- current of 100 μA, decreases by 50% over a time of about 2000 h. For this spectral range, this is a good indicator, but for the final conclusions, additional studies are required for various types of the MQW structures in an (Al,Ga)N system.

In summary, UVC emitters with CNT-based e-source typically have large light-emitting surface diameters of up to 2 inches, a relatively low e-beam energy (<10 keV) and a current of few mA. At the same time, these emitters have the highest WPE (3–5.25%) [71,122]. Therefore, these UVC emitters can safely operate in cw- mode with a maximum output optical power up to 702 mW and a relatively long lifetime. On the other hand, only UVC emitters pumped by pulsed high-current (up to 2A) e-beam guns can achieve the highest peak optical powers up to 50 W [125]. However, the lifetime of these emitters will be relatively short due to the degradation of both the e-source and the light-emitting structure. Thus, despite the lack of systematic studies, it is possible to expand the boundaries of the damage-free use of e-beam-pumped UVC emitters by appropriately choosing their type and operating mode. The qualitative nature of these conclusions indicates the need for research and development in this area.

## 7. Conclusions

Electron-beam-pumped light emitters operating in the UVC wavelength range (220–280 nm) and especially in its far-UVC subrange (~230 nm) are an attractive alternative to UVC-LEDs, as they eliminate the critical problem of acceptor and donor doping of Al-rich AlGaN layers and significantly increase the output optical power of a single device. These devices can be used in equipment for the environmentally friendly and safe optical disinfection of air/water/surfaces, in spectroscopic devices for highly sensitive detecting various substances, in NLOS communication systems, etc. 

At present, various types of e-guns have been developed with varying degrees of industrial mass production, ranging from the most common e-guns with thermionic cathodes (W- or LaB_6_-based) to modern e-emitters using either large-area CNT-based field emission cathodes or high-current plasma cathodes of various types. The typical AlGaN-based MQW heterostructures pumped by an e-beam demonstrate UVC emissions in the spectral range of 240–280 nm.

Using thermionic e-guns with conventional tungsten cathodes operated at a typical anode voltage of 10–20 kV and e-beam currents of a few mA, the output optical power of UVC radiation can reach several hundred milliwatts in pulsed excitation mode and several tens of milliwatts in cw- mode. The efficiency of excitation of UVC radiation does not exceed 1%.

An increase in the e-beam current up to several tens of milliamps is possible for e-guns with thermionic cathodes based on hexaboride compounds of rare earth metals (LaB_6_, etc.), operating in pulsed mode. The UVC emitters with these e-guns achieve peak output optical powers of up to 1 and 2 W with wavelengths of around 240 and 258 nm, respectively. However, the excitation efficiency in these UVC emitters remains extremely low (<<1%).

Significantly higher excitation efficiency up to 5.2% is achieved in AlGaN-MQW UVC emitters pumped by e-guns with cold field emission cathodes based on graphene nanotips or CNTs deposited or formed on large-area substrates of up to 2 inches in diameter. The output optical power of these UVC emitters can reach 700 mW at 248 nm when cw-excited by a CNT e-gun with an operating voltage of less than 10 kV and a current of about 1 mA, which can be provided by compact power supplies.

The maximum peak output optical power up to several tens of watts is observed in UVC emitters pumped by a high-current pulsed e-beam (up to 2 A) generated by an e-gun with a plasma ferroelectric cathode. However, in this case, the quantum efficiency does not exceed 1%.

Most e-beam-pumped UVC emitters used MQW structures in an (Al,Ga)N system, grown using MOCVD or PA MBE. Removing the requirement for the conductivity of light-emitting heterostructures in e-beam-pumped UVC emitters allows the use of Al_x_Ga_1−x_N/AlN heterostructures or even those based on GaN/AlN binary alloys. In the latter case, for a short-wavelength shift of the output radiation to the desired UVC range of 230–260 nm, it is necessary to reduce the QW thickness to 1–2 ML, which is currently provided by both PA MBE and MOCVD technologies. Moreover, the ML range of QW thicknesses leads to a significant weakening of the quantum-confined Stark effect and the elimination of harmful switching from the TE- to TM-polarized mode of the output radiation. Both effects play a significant role in reducing the quantum efficiency and light extraction efficiency in AlGaN-based MQW structures with a high Al content and conventional thickness of 1–2 nm.

At present, the maximum values of the internal quantum yield and output optical power have been achieved for structures with a nominal QW thickness of about 1.5 ML, a number of QWs up to 400 and a total thickness of the MQW structure up to 2 µm. In addition, the possibility of a controlled change in the emission wavelength in the range from 238 to 265 nm with maximum output optical powers of 10 and 50 W, respectively, was demonstrated for the structures grown on *c*-Al_2_O_3_ substrates. It is also important that similar structures exhibit linear non-saturated power dependences on the pulsed pump current up to maximum values.

It should be noted that there is an urgent need to test the lifetime of various e-beam-pumped UVC emitters and to conduct more thorough studies om the degradation processes in (Al,Ga)N-based UVC-emitting heterostructures and e-beam sources.

Thus, it can be expected that the production of e-beam-pumped UVC emitters with a high output optical power and high efficiency in the entire wavelength range of 220–270 nm to meet current demand will be ensured in the near future.

## Figures and Tables

**Figure 1 nanomaterials-13-02080-f001:**
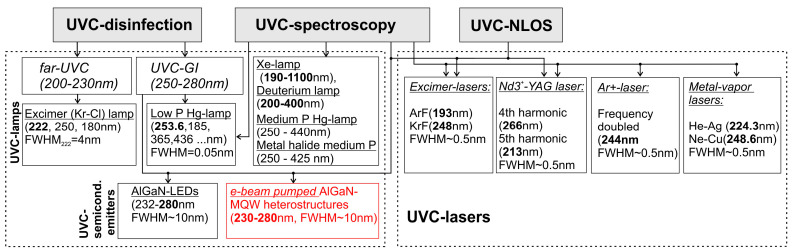
Main areas of application of modern lamp-based UVC emitters and the developed semiconductor UVC-LEDs and e-beam-pumped UVC emitters.

**Figure 2 nanomaterials-13-02080-f002:**
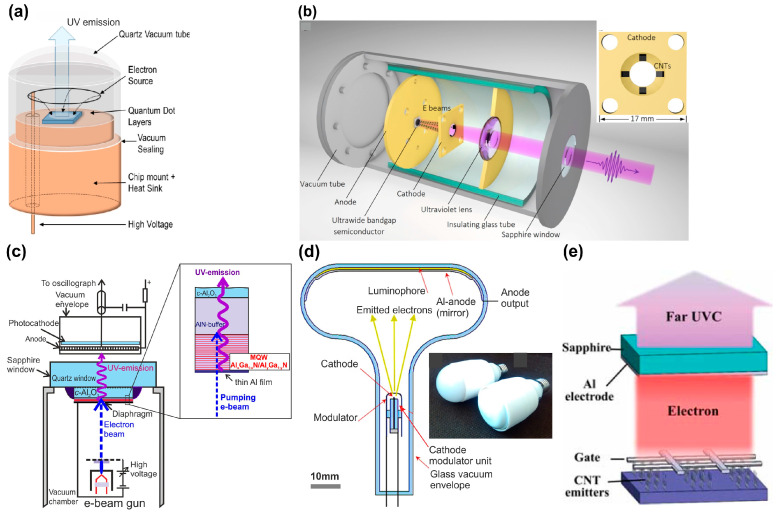
E-beam lamps of type I with a grid cathode (**a**) [54] and a four-segmented CNT cathode (**b**) [55]. The inset in (**b**) shows a photo of the latter cathode. E-beam lamps of type II scheme for measuring the characteristics of an e-beam-pumped UVC emitter using a thermionic e-gun and a photocathode for the optical measurements (**c**) [56]. The inset shows an excitation scheme of an (Al,Ga)N-based MQW heterostructure. E-beam lamps with a built-in e-gun and a voltage converter for the E27 standard (**d**) [57]. Schematic of excitation of a UVC emitter using a field emission e-gun based on large-area CNT cathode (**e**) [58].

**Figure 3 nanomaterials-13-02080-f003:**
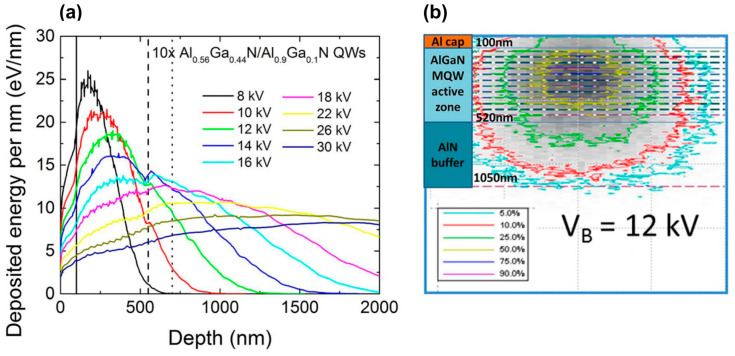
(**a**) Results of Monte Carlo calculations of absorption distribution profiles for e-beam energies with energies from 8 to 30 keV. The vertical dotted lines show the boundaries of the 10 × {Al_0.56_Ga_0.44_N/Al_0.9_Ga_0.1_N} light-emitting MQW region with a length of 150 nm in the AlN layer of the UVC emitter. (**b**) Two-dimensional absorption distribution of a point e-beam with an energy of 12 keV. The area of maximum excitation (50%) is marked with a yellow contour in the MQW structure [60].

**Figure 4 nanomaterials-13-02080-f004:**
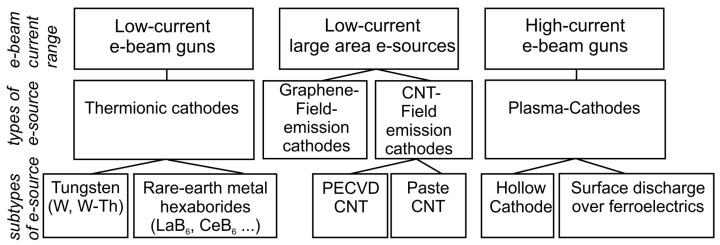
Main types of electron sources for pumping (Al,Ga)N-based UVC emitters.

**Figure 5 nanomaterials-13-02080-f005:**
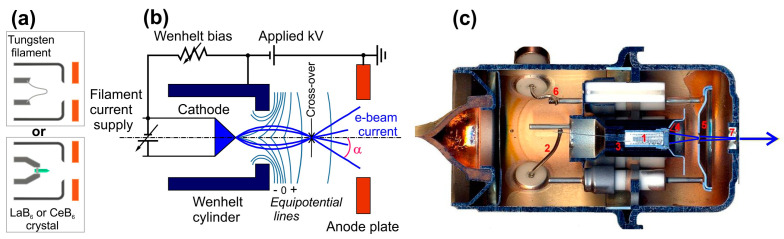
(**a**) Schemes of cathode assemblies of various types: tungsten (upper) or rare-earth metal hexaborides (lower). (**b**) General scheme of a thermionic e-beam gun. (**c**) Cross-sectional photograph of a thermionic e-beam gun with an incandescent cathode: 1—heated element, 2, 3—wires to the heater, 4—Wehnelt cylinder, 5—anode, 6—high voltage contact, 7—e-beam output.

**Figure 6 nanomaterials-13-02080-f006:**
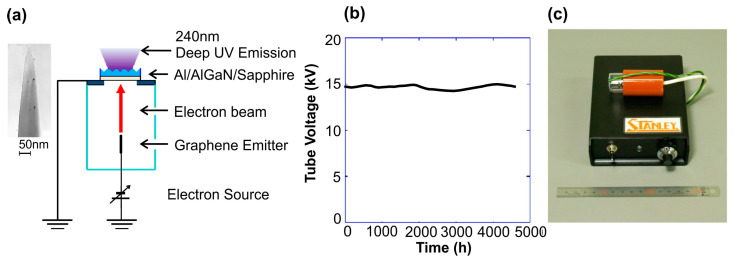
(**a**) Design of an e-gun with a field (graphene) cathode, the TEM image of which is shown in the inset. (**b**) Change in cathode voltage with time at a current of 500 µA. (**c**) Photograph of a UVC emitter with a compact cylindrical e-beam gun and a portable power supply that provides adjustment of the e-beam energy in the range of 0–10 keV and its current of 0–500 μA [66].

**Figure 7 nanomaterials-13-02080-f007:**
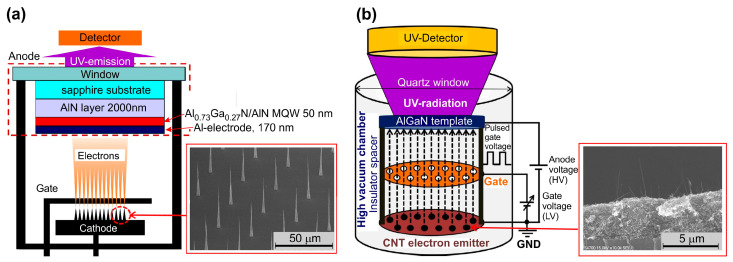
Triode schemes of e-beam guns with CNT e-emitters (cathodes) of various types, formed (**a**) by photolithography as regularly arranged nano-tips [70]; (**b**) by a multistep process of mixing CNT powder and organic solvents followed by annealing, adhesive type and soft roller treatment, which led to the random formation of vertical CNTs [69]. The insets in each figure show SEM images of the corresponding CNT e-emitters.

**Figure 8 nanomaterials-13-02080-f008:**
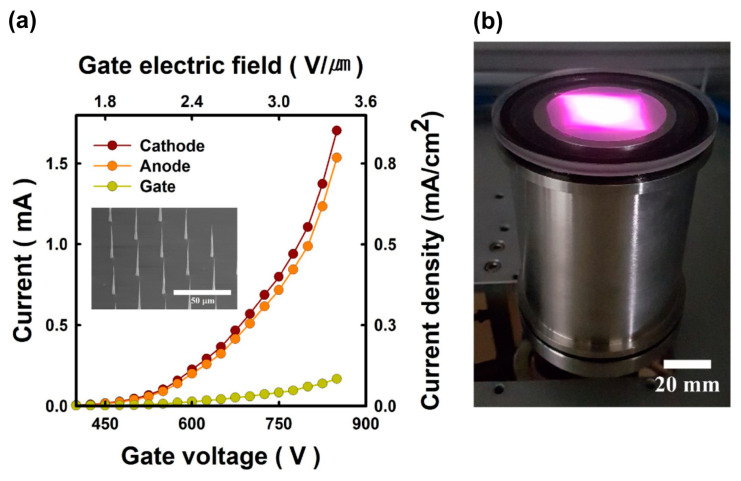
(**a**) Current–voltage characteristics of anode, cathode and gate e-beam currents measured in a nanotip CNT e-emitter. (**b**) Photo of UVC emitter with this e-beam CNT emitter [50].

**Figure 9 nanomaterials-13-02080-f009:**
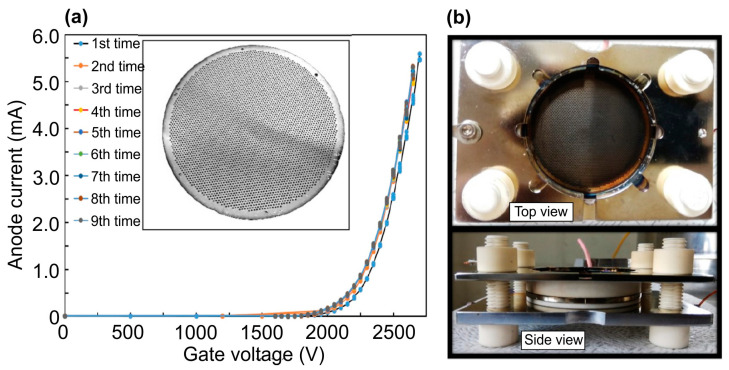
(**a**) Current–gate voltage characteristics of two-inch CNT e-emitters. The inset shows an image of the on a stainless-steel substrate for CNT deposition. (**b**) Plan- and side-view photos of the setup for CL measurements from the assembly of the UVC emitter with a CNT e-emitter, the triode scheme of which is shown in Figure 7b [71].

**Figure 10 nanomaterials-13-02080-f010:**
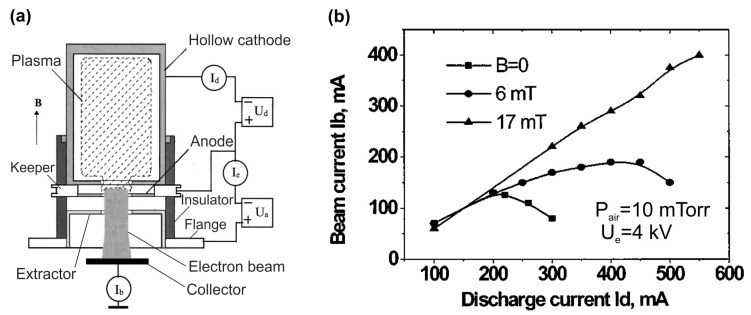
Scheme of an e-gun with a hollow plasma cathode (**a**) and the dependence of the current of the e-beam on the discharge cw- current at a different axial magnetic field in a hollow plasma cathode (**b**) [73].

**Figure 11 nanomaterials-13-02080-f011:**
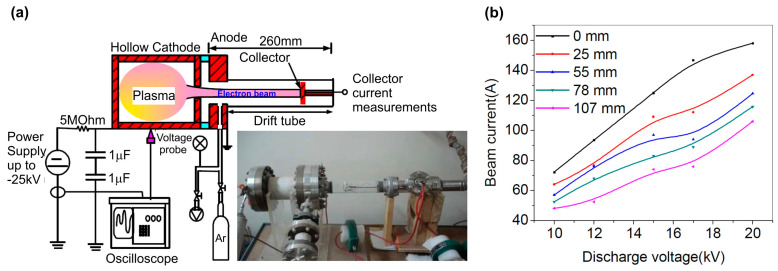
Scheme and photo of a pulsed pseudo-spark e-gun with a hollow plasma cathode (**a**) and the dependence of the peak e-beam current on voltage at different locations of the drift space (**b**) [74].

**Figure 12 nanomaterials-13-02080-f012:**
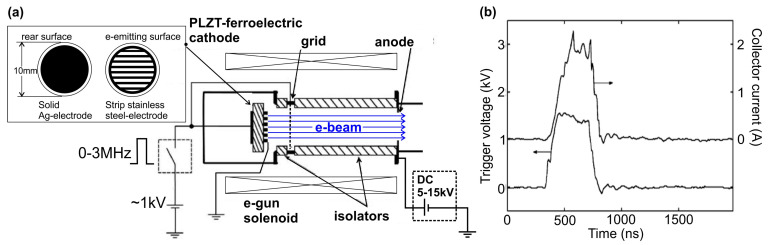
(**a**) Scheme of an e-source with a plasma cathode based on a surface discharge over a ferroelectric cathode. The inset shows the metal electrodes on the rear and e-emitting surfaces of the ferroelectric cathode. (**b**) Time dependence of the output e-current from such a source [80].

**Figure 13 nanomaterials-13-02080-f013:**
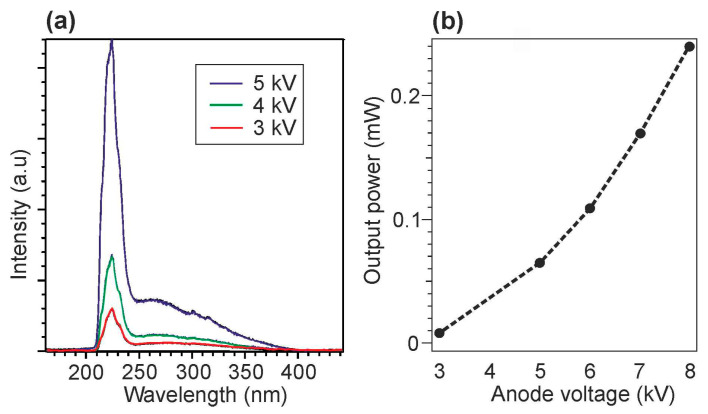
(**a**) CL-RT spectra emitted by pyro-BN powder pumped with an e-beam with different electron energies, which demonstrate a main peak at 225 nm, the broadening of which is explained by the superposition of exciton levels in a range from 215 to 227 nm. The low-intensity long-wavelength tail in the spectra from 250 to 400 nm is explained by emissions from impurities and defects in the pyro-BN material. (**b**) Dependence of CL output power on the accelerating voltage in the e-gun from 3 to 8 kV, which changes the power value up to ~0.2 mW [87].

**Figure 14 nanomaterials-13-02080-f014:**
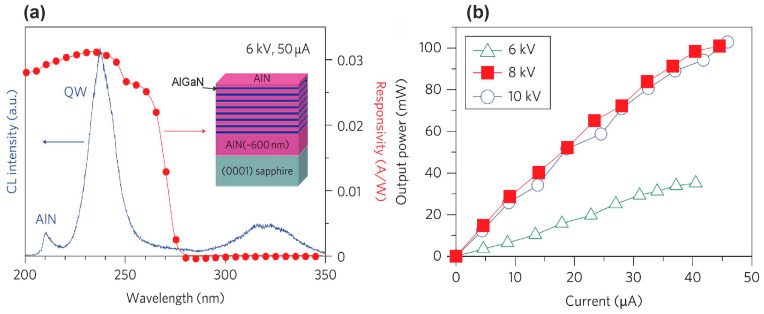
(**a**) CL-RT spectrum observed when the Al0.69Ga0.31N/AlN MQW structure was excited by an e-beam with an energy of 6 keV and a current of 50 µA. The figure also shows the spectral sensitivity of the UV photodiode used in the measurements. The inset shows a scheme of the MQW structure under study. (**b**) Dependences of output power of the e-beam-pumped UVC emitter on the e-beam current with different energies [88].

**Figure 15 nanomaterials-13-02080-f015:**
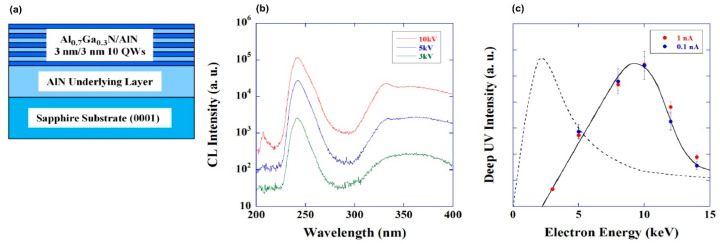
(**a**) Schematic of the 10 × {Al_0.7_Ga_0.3_N(3 nm)/AlN(3 nm)} MQW heterostructure; (**b**) CL-RT spectra of this MQW heterostructure at different electron energies, (**c**) Theoretical (dashed line) and experimental dependences (solid line) of CL intensity on the electron energy [66].

**Figure 16 nanomaterials-13-02080-f016:**
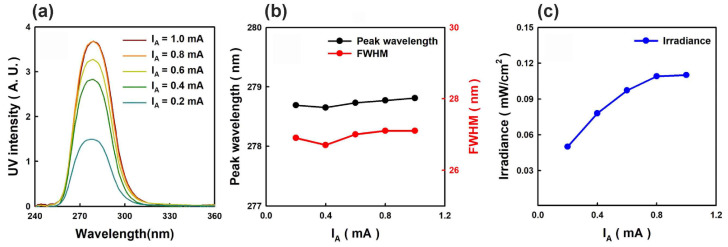
Characteristics of the 5 × {Al_0.47_Ga_0.53_N/Al_0.56_Ga_0.44_N:Si} MQW structure with a light-emitting area of 303 mm^2^: (**a**) CL-RT spectra measured at an energy of 3 keV and various e-beam currents of 0.2–1.0 mA; (**b**) The variations of CL peak wavelength and its full width at half maximum (FWHM) on an e-beam current; (**c**) Dependence of output optical power on the e-beam current [68].

**Figure 17 nanomaterials-13-02080-f017:**
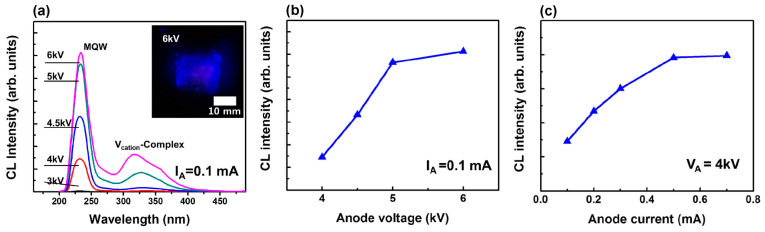
(**a**) CL spectra of the Al_0.73_Ga_0.27_N(2 nm)/AlN(8 nm) MQW heterostructure pumped over an area of 276 mm^2^ at various e-energies in the range of 3–6 keV, emitted from an e-source with nanotip CNT cathodes. The inset shows a photo of radiation in the visible range. The dependences of CL-RT intensity on the e-beam energy and the anode current are shown in Figures (**b**,**c**), respectively [70].

**Figure 18 nanomaterials-13-02080-f018:**
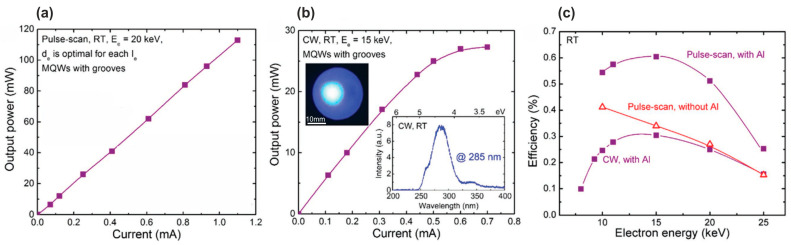
Dependences of output optical powers of 40 × {3 × [(GaN)_1_/(Al_0.75_Ga_0.25_N)_2_]/Al_0.75_Ga_0.25_N(31 nm)} MQW structure on the e-beam current excited in pulsed-scan (**a**) and cw- (**b**) pumping modes with an e-beam energy of 20 and 15 keV, respectively. The upper inset in (**b**) shows the emission through paper irradiated by an MQW structure excited by an e-beam with an 8 mm beam spot diameter, and the lower inset shows the CL spectrum. (**c**) Dependences of WPE at RT of AlGaN-based UVC emitters on the e-beam energy in different pumping modes [104].

**Figure 19 nanomaterials-13-02080-f019:**
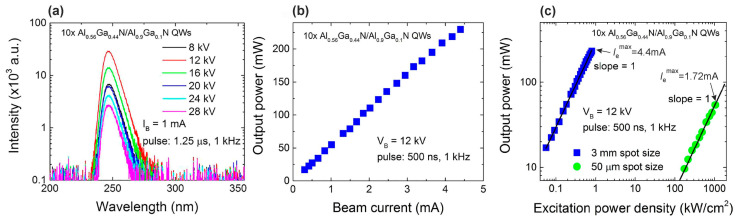
(**a**) CL-RT spectra of the 10 × {Al_0.56_Ga_0.44_N/Al_0.9_Ga_0.1_N} MQW structure, measured at e-beam energies varying from 8 to 28 keV and an e-beam current of 1 mA with a pump pulse duration of 1.25 µs and a frequency step of 1 kHz. (**b**) Dependence of output optical power on the pump current by electrons with an energy of 12 keV, a pulse duration of 500 ns and a repetition rate of 1 kHz. (**c**) Comparison in a double-logarithmic plot of the CL output power vs. the e-beam excitation power density for the incident e-beam diameters of 3 mm and 50 µm [60].

**Figure 20 nanomaterials-13-02080-f020:**
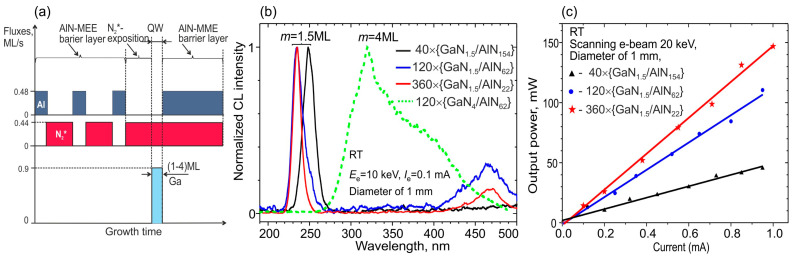
(**a**) Typical shutter sequence during the growth of a GaN/AlN QW with a various well thickness determined by the Ga exposure time. (**b**) Normalized RT CL spectra of the *N* × {GaN_1.5_/AlN_m_} MQW structures with *N* = 40–360 and *m* = 22−154, measured at *E*_e_ = 10 keV, *I*_e_ = 0.1 mA and a beam spot diameter of ∼1 mm. (**c**) Peak output power as a function of the e-current of the pulse-scanning e-beam with *E*_e_ = 20 keV, measured at RT for the MQW structures [110].

**Figure 21 nanomaterials-13-02080-f021:**
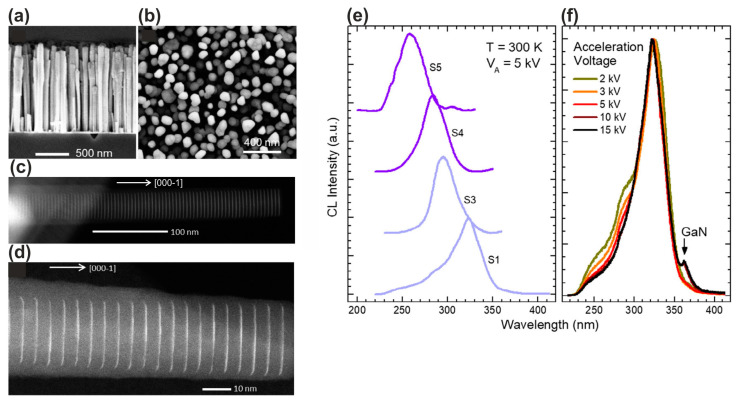
Cross-section (**a**) and plan-view (**b**) SEM images of a nanowire structure with 88 × {Al_0.1_Ga_0.9_N/AlN} MQW [112]. HAADF STEM images of dispersed NWs containing 88 × {GaN(1.5 nm)/AlN(3 nm)} (**c**) and 88 × {(Al_0.1_Ga_0.9_N)(0.65 nm)/AlN(3.85 nm)} (**d**). (**e**) Normalized CL-RT spectra for the following structures: S1: 88 × {GaN(1.5 nm)/AlN(3.0 nm)}; S3: 88 × {GaN(0.75 nm)/AlN(3.75 nm)}; S4: 88 × {(Al_0.1_Ga_0.9_N)(0.75 nm)/AlN(3.75 nm)}; and S5: 88 × {(Al_0.1_Ga_0.9_N)(0.65 nm)/AlN(3.85 nm)}. For clarity, the spectra are shifted vertically [112]. (**f**) Normalized CL spectra of the MQW structure S4: 88 × {(Al_0.1_Ga_0.9_N)(0.75 nm)/AlN(3.75 nm)}, measured at various electron energies from 2 to 15 kV. The CL peak associated with the emission from GaN in the bases of nanocolumns is observed only at an accelerating voltage above 5 kV [112].

**Figure 22 nanomaterials-13-02080-f022:**
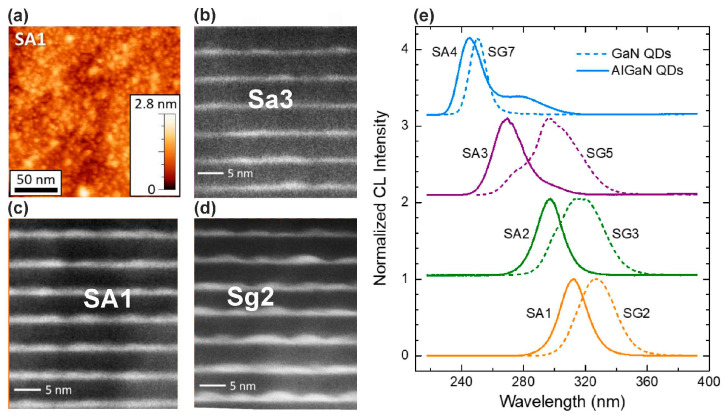
(**a**) AFM image of the surface of the 100 × {(Al_0.1_Ga_0.9_N)_5_/AlN(~2 nm)} structure. HAADF STEM images of the following MQW structures: SA3:100 × {(Al_0.1_Ga_0.9_N)_2.9_/AlN(~2 nm)} (**b**), SA1: 100 × {(Al_0.1_Ga_0.9_N)_5_/AlN(~2 nm)} (**c**) and SG2: 100 × {GaN_4.6_/AlN(~2 nm)} (**d**). These structures were grown under nitrogen-enriched conditions with flux ratios of III/N = 0.47(SA3), 0.8(SA1), and 0.73(SG2) [114]. (**e**) Normalized CL-RT spectra for GaN/AlN (SG) (dashed lines) and Al_0.1_Ga_0.9_N/AlN (SA) (solid lines) MQW structures grown with various *m* and Ga fluxes: SA4, SG7—*m* = 1.8–2.0 ML, Ga = 0.149 ML/s; SA3, SG5—*m* = 2.6–2.9 ML, Ga = 0.220 ML/s; SA2, SG3—*m* = 3.8–4.2 ML, Ga = 0.319 ML/s; and SA1, SG2—*m* = 4.6–5 ML, Ga = 0.38 ML/s (from top to bottom, respectively) [114], SA4, SG7(blue)—*m* = 1.8–2.0 ML, Ga = 0.149 ML/s; SA3, SG5(purple)—*m* = 2.6–2.9 ML, Ga = 0.220 ML/s; SA2, SG3(green)—*m* = 3.8–4.2 ML, Ga = 0.319 ML/s; and SA1, SG2(orange).

**Figure 23 nanomaterials-13-02080-f023:**
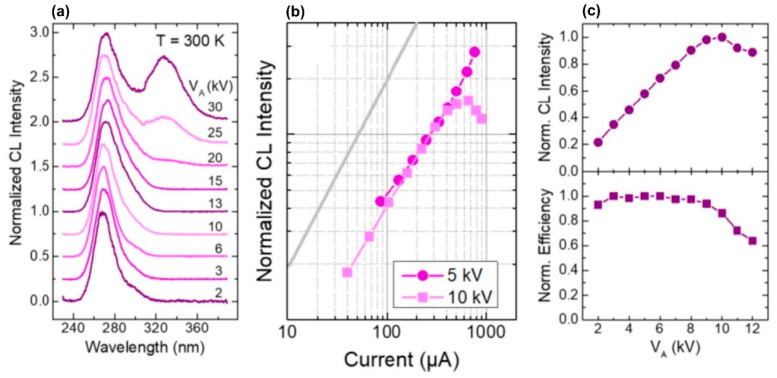
(**a**) Normalized CL spectra of the 100 × {(Al_0.1_Ga_0.9_N)_2.9_/AlN(~2 nm)} MQW structure, excited at various e-beam energies. (**b**) Variation of CL intensity as a function of the injection current measured for the same sample at *E*_e_ = 5 keV and 10 keV. The slope of the solid grey line corresponds to a linear increase. No saturation is observed up until 800 μA for *E*_e_ = 5 keV while a saturation for currents higher than 400 μA is observed at *E*_e_ = 10 keV. (**c**) Normalized CL intensity (upper curve) and emission efficiency (lower curve) as a function of an anode voltage for the same sample. Measurements were performed using an e-gun operated in direct current mode, under normal incidence, with a beam spot diameter of 4 ± 1 mm [114].

**Figure 24 nanomaterials-13-02080-f024:**
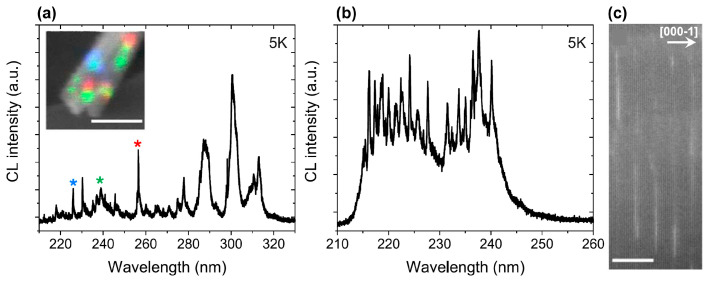
(**a**) 5K CL spectrum of two single NWs shown in the inset. The color code refers to different spots emitting at different wavelengths. The peak at 240 nm is assigned to 1 ML thick GaN QWs. The scale bar in the inset is 200 nm. (**b**) 5K CL spectrum in another spot, emphasizing the presence of sharp lines below 240 nm, which are assigned to the presence of incomplete QWs. (**c**) STEM-HAADF image of the incomplete GaN QWs. The scale bar is 5 nm [115].

**Figure 25 nanomaterials-13-02080-f025:**
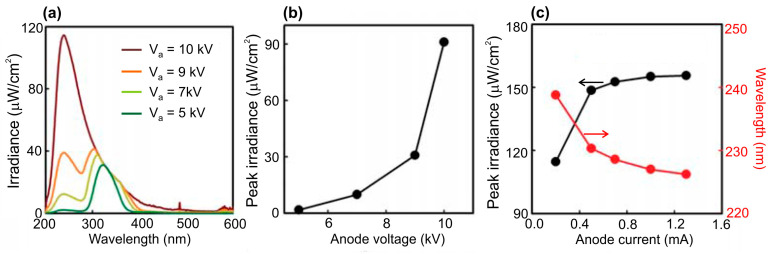
(**a**) Spectral density of output CL power from a sapphire excited by e-beams with various electron energies in the range of 5–10 keV, (**b**) dependence of the peak density of output optical power on the anode voltage (e-beam energy), (**c**) dependences of CL peak spectral position and density of output peak optical power on the anode current [58].

**Figure 26 nanomaterials-13-02080-f026:**
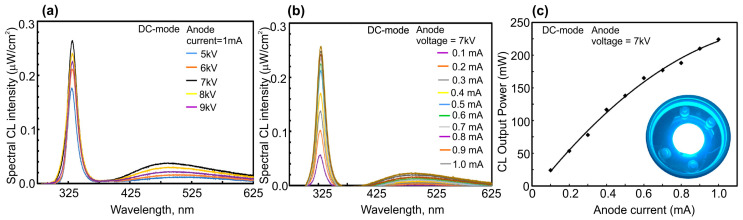
CL-RT spectra of an 18 × {GaN(2.1 nm)/AlGaN(9 nm)} MQW structure, measured at various e-beam energies (**a**) and currents (**b**). (**c**) Dependence of the CL output power from this structure on the e-beam current at an anode voltage of 7 kV. The inset shows a photo captured during actual UV–CL emission from the two-inch AlGaN/GaN MQW wafer at an anode voltage of 7 kV and anode current of 1 mA under the DC electric field [71].

**Figure 27 nanomaterials-13-02080-f027:**
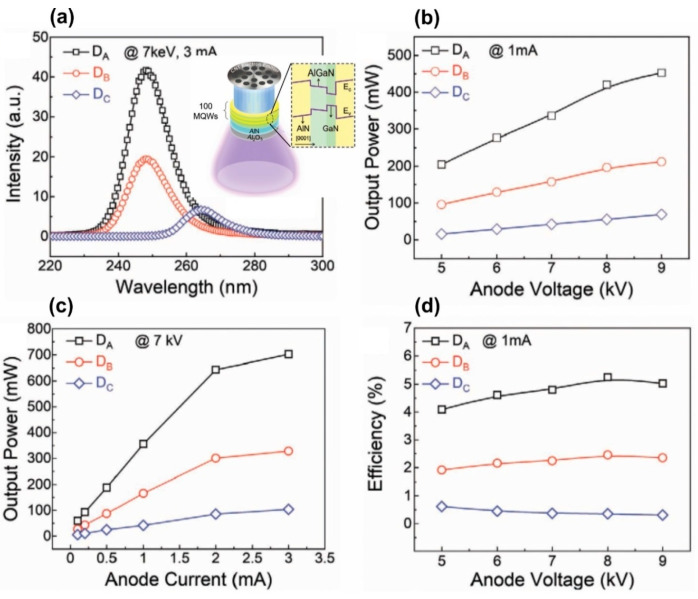
(**a**) CL spectra of three 100 × {GaN_1_/(Al_x_Ga_1−x_N)_2_/AlN_40_} MQW structures with various Al—contents in the 2 ML thick Al_x_Ga_1−x_N layer: *x* = 0.6(D_A_), 0.5(D_B_), 0(D_C_). Dependences of output power of these three structures on the anode voltage (**b**) and anode current (**c**). (**d**) Dependences of WPE on the anode voltage for these three structures [122].

**Figure 28 nanomaterials-13-02080-f028:**
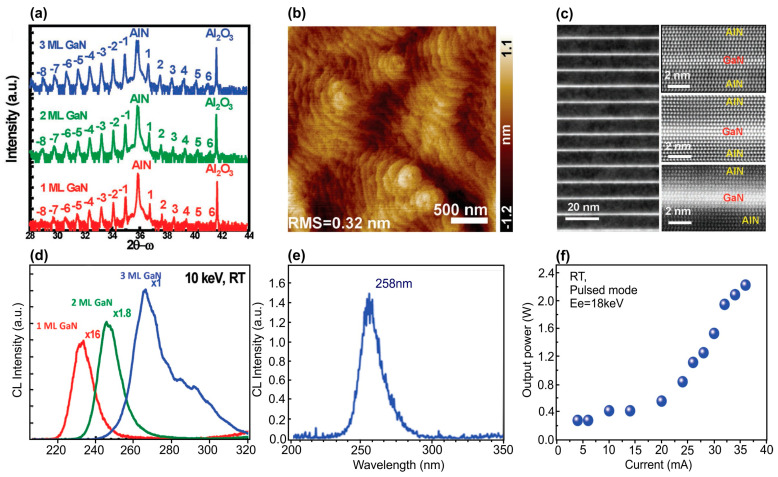
(**a**) XRD ω-2θ scans of the symmetric (0002) plane for samples with a one ML GaN well (red line), two ML GaN wells (green line), and three ML GaN wells (blue line). (**b**) AFM image taken in a scanned area of 3 × 3 μm^2^ with an RMS value of 0.32 nm. (**c**) Cross-sectional HAADF-STEM images in the region of several periods of a nominal one ML GaN (left) and one period of one, two, and three ML GaN wells (right). (**d**) CL spectra of 100 × {GaN_m_/AlN_40_} MQW structures with different nominal well thicknesses (*m* = 1, 2, 3), grown with MOCVD. (**e**) CL spectrum of the 150 × {GaN_2_/AlN_40_} MQW structure with optimized parameters (**f**) and its output optical power via the e-current supplied by an e-gun with a LaB_6_ thermionic cathode operated in pulsed mode [123].

**Figure 29 nanomaterials-13-02080-f029:**
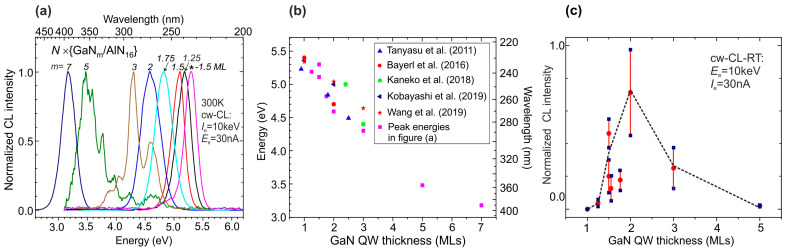
(**a**) Normalized CL-RT spectra of *N* × {GaN_m_/AlN_16_} MQW structures, measured at an e-beam energy and current of 10 keV and 30 nA, respectively. Structure grown at *m* = 7 ML had *N* = 25, while all others with *m* = 1.25–5 were grown with N = 100. All structures were grown in one series, as described in [124]. The spectrum marked by «*» was measured for the structure grown at *m* = 1.5 ML, *n* = 22 ML and *N* = 120, as described in [110]. (**b**) Literature review of the dependence of the energy positions of CL and PL peaks on the nominal QW thickness in MQW structures [106,107,123,128,129]. (**c**) Dependence of the CL intensity on the QW thickness in the MQW structures whose spectra are shown in (**a**) [124].

**Figure 30 nanomaterials-13-02080-f030:**
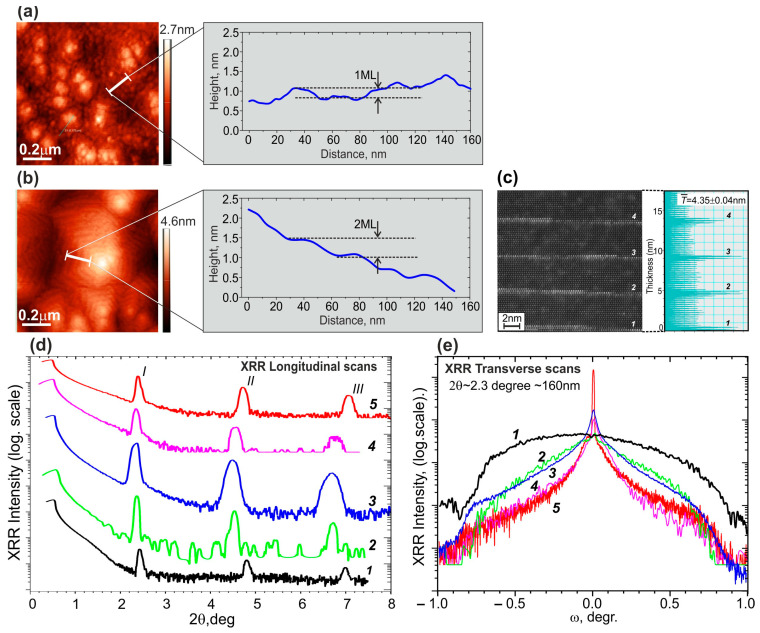
AFM images of the surfaces of 400 × {GaN_1.5_/AlN_16_} MQW structures with QWs grown at Ga/N_2_* ratios of 1.1 (**a**) and 2.2 (**b**). Insets in the figures depict changes in the profile height along the white line segments plotted on these figures. (**c**) Cross-sectional HAADF STEM image of the 100 × {GaN_1.5_/AlN_16_} MQW structure with QWs grown at Ga/N_2_* of 2.2. The inset shows integrated brightness distributions in the vertical direction. The longitudinal (Bragg-like) 2θ- (**d**) and transverse ω- (**e**) scans of X-ray reflectivity curves of five 400 × {GaNm/AlN16} MQW structures 1–5 with various nominal well thicknesses of m: (1: m = 1.2; (2–5): m = 1.5) grown at different Ga/N2* flux ratios on different AlN/c-Al2O3 templates: 1: Ga/N2* = 0.6 on PA MBE-AlN template; 2, 3: Ga/N2* = 2.2, 4, 5: Ga/N2* = 1.1; 3, 4—on AlN-PA MBE templates; and 2, 5—on double MOCVD/PAMBE ones. I, II and III in (**d**) indicate the positions of the three first Bragg peaks [125].

**Figure 31 nanomaterials-13-02080-f031:**
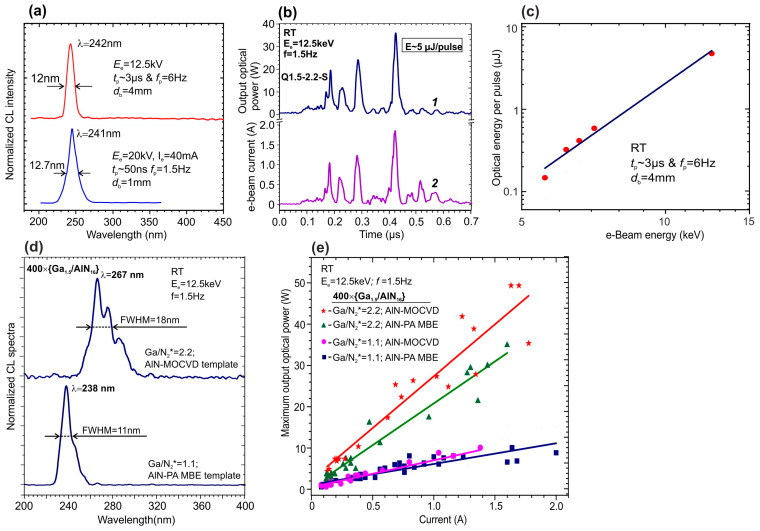
(**a**) Normalized CL-RT spectra of the {GaN_1.5_/AlN_22_} MQW structures, measured using various e-guns with a LaB_6_ cathode (lower blue spectrum) and a plasma ferroelectric cathode (upper red one) [124]. (**b**) The output optical power waveform (upper, blue) from the 360 × {GaN_1.5_/AlN_22_} MQW structure excited by an e-beam current at *E*_e_ = 12.5 keV, with the waveform shown in the lower purple curve [125]. (**c**) Dependence of output UVC-optical emitted by 400 × {GaN_1.5_/AlN_22_} MQW structure per excitation pulse on the voltage applied to a plasma ferroelectric cathode of the e-gun [124]. (**d**) Normalized CL-RT spectra excited with a pulse e-gun with a plasma cathode at *E*_e_ = 12.5 keV, which were measured in 400 × {GaN_1.5_/AlN_16_} MQW structures grown under various Ga/N_2_* flux ratios and different AlN templates. (**e**) Dependences of output optical power on the current of an energy of 12.5 keV, measured for different 400 × {GaN_1.5_/AlN_16_} MQW structures. The lines are drawn in accordance with the linear interpolation of the experimental data [125].

**Table 1 nanomaterials-13-02080-t001:** Main parameters and characteristics of the principal e-beam sources.

	Units	Thermionic Cathodes		Carbon-Based Cold Field Emitters			Plasma Cathodes	
		W	LaB_6_	Graphene Nanoneedle	CNT-CVD	CNT-Paste	Hollow Cathode	Ferroelectric Cathode
Work function, (W)	eV	4.5	2.4–2.7	-	5	-	-	-
Richardson’s constant, (A)	A·m^−2^K^−2^	6 × 10^9^	4 × 10^9^	-	-	-	-	-
Operating temperature, (T)	K	2700	1700	300	300	300	300	300
Crossover (beam) size	mm	>0.1	0.01	-	-	-		
Brightness	A·m^−2^sr^−1^	10^10^	5 × 10^11^	10^13^				
Emission current stability	% hr^−1^	<1	<1	good				poor
Vacuum	Pa	10^−2^	10^−4^	10^−4^	10^−5^	10^−5^	10^−1^	10^−2^–10^−3^
Lifetime	hr	100	1000	>5000				>10^7^ pulses
Gate voltage	kV	-	-	-	~1.5	2.7		0.8–1.5
Anode voltage	kV	10–20	10–20	3–10	~5	7		5–15
Maximum anode (e-beam) current	A	1–5	60	0.5	1.5	~5.5	cw: 0.4 peak: 120	peak: 2
Technology				H_2_-plasma etching	PECVD	MOCVD+ screen- printing		
Radius of tip	nm	-	-	5	60	5–7	-	-
e-emission area	mm^2^	~1	~1	-	~300	Up to 2-inch	>250	~20
Operation mode		Cw- and Pulsed	Pulsed	Cw- and Pulsed	Cw- and Pulsed	Cw- and Pulsed	Cw- and Pulsed	Pulsed

**Table 2 nanomaterials-13-02080-t002:** Main results for e-beam-pumped UVC emitters.

No	Emitter (Powder/Layer/ MQW Structure)	Technology	λ	Output Optical Power		e-Beam				WPE (IQE)	Life Time	Year, Institute	Ref.
				Pulse (Scan)	cw-	Cathode	Energy	Current	Area				
			nm	mW	mW		keV	mA	cm^2^	%	hrs		
1	Pyro-BN Powder	HP HT	225		0.2	W-TE	8	0.05	0.2	0.5	>150	2009, National Institute for Materials Science	[87]
2	8 × {Al_0.69_Ga_0.31_N(1 nm)}/ AlN(15 nm)	MO CVD	240		100	W-TE	8	0.05	0.002	40	1	2010, Kyoto Univ.	[88]
3	AlGaN:Si(800 nm)/AlN Layer	MO CVD	247		2.2	TE	10	0.1	0.003	0.24	>2 × 10^3^	2011, Mie Univ. Hamamatsu	[89]
4	10 × {Al_0.7_Ga_0.3_N(3 nm)/ AlN(3hm)	MO CVD	240		20	Graphe- ne -FE	7.5	0.08	7	3–4	>5 × 10^3^	2012, Stanley Electric corp. and Nagoya Univ.	[66]
5	70 × Al_0.6_Ga_0.4_N:Si(1.5 nm)/ Al_0.75_Ga_0.25_N:Si(7 nm)	MO CVD	256		15	TE	10	0.2	0.002	0.75	-	2013, Mie Univ. and Hamamatsu	[90]
6	40 × {3 × [(GaN)_1_/ (Al_0.75_Ga_0.25_N)_2_]/ (Al_0.75_Ga_0.25_N)_124_}	PA MBE	270	60	4.7	W-TE	ps: 32p cw: 20	ps: 1.2 cw: 0.1	0.5	0.19 0.24	-	2015, IOFFE	[102]
7	40 × {3 × [(GaN)_1_/ (Al_0.75_Ga_0.25_N)_2_]/ (Al_0.75_Ga_0.25_N)_124_}	PA MBE	285	160	39	W-TE	ps: 20 cw: 15	ps: 1.1 cw: 0.7	ps: 3 × 10^−4^ cw: 0.8	ps: 0.86 cw: 0.43	-	2016, Peking Univ. and IOFFE	[104]
8	10 × Al_0.56_Ga_0.44_N(1.5 nm)/ Al_0.9_Ga_0.1_N(40 nm)	MO CVD	246	230	-	W-TE	p: 12	p: 4.4	p: 0.07	p: 0.43	-	2016, Palo-Alto and TU-Berlin	[60]
9	360 × {GaN_1.5_/AlN_22_}	PA MBE	235	150	28	W-TE	ps: 20 cw: 15	ps: 1.0 cw: 0.45		ps: 0.75 cw: 0.42		2018, IOFFE	[110]
10	5 × Al_0.47_Ga_0.53_N/ Al_0.56_Ga_0.44_N	MO CVD	279	-	30	CNT	3	0.8	3.03	1.25		2019, Kying Univ. Korea	[68]
11	100 × {GaN_3_/AlN_40_} 100 × {GaN_2_/AlN_40_} 100 × {GaN_1_/AlN_40_} 150 × {GaN_2_/AlN_40_}	MO CVD	265 243 232 258	179 123 25 2200	53 39 0.8	TE TE TE LaB_6_	Ps:20 Ps:20 Ps:20 Ps:18	1.2 1.2 1.0 Ps: 37	0.01 0.01 0.01 0.01	0.75 0.51 0.13 0.33		2019, Peking Univ. and IOFFE	[123]
12	Al_0.73_Ga_0.27_N/AlN	MO CVD	233	-	6.4	CNT	4	0.5	2.83	0.32	-	2020, Kying Univ. Korea	[70]
13	NW-88 × {(Al_x_Ga_1−x_N) (0.65-1.5hm) (x = 0 or 0.1)/ AlN (3–4hm)	PA MBE	258– 340	-	-	W-TE	2-15	<120 pA	-	IQE: 22–63%	-	2020, Univ. Grenoble-Alpes	[113]
14.	NW-88 × {(Al_x_Ga_1−x_N) (0.65–1.5hm) (0 ≤ x ≤ 0.1)/ AlN (3-4hm)	PA MBE	258– 340				3-10	0.4	0.13 (4 mm Dia)	IQE: >60%		2020, Univ. Grenoble-Alpes	[114]
15	100 × {QD-QW(Al_x_Ga_1−x_N) (0.65-1.5hm) (x = 0 or 0.1)/ AlN(3–4hm)	PA MBE	244– 335	-		W-TE	3–10	0.8	0.13 (4 mm Dia)	IQE: 33–54%		2020, Univ. Grenoble-Alpes	[115]
16.	Bulk Al_2_O_3_ wafer		226–400	-	Peak: 1.5	CNT	10	1.3	9.6	Peak: 0.01		2020, Kying Univ. Korea	[69]
17.	18 × {GaN(2.1 nm)/ Al_0.36_Ga_0.64_N(9 nm)}	MO CVD	330		225	CNT	7	1	~20	3.6		2021, Chonnam Nat. Univ.	[71]
18	400 × {GaN_1.5_/AlN_16_}	PA MBE	240 242	10^3^ (11.8 × 10^3^) 5 µJ/ Pulse	-	LaB_6_ TE PC	P: 20 P: 12.5	P: 65 P: 450	0.008 (1 mm) 0.13 (4 mm Dia)	0.08 0.2		2021, IOFFE and Peking Univ.	[124]
19	100 × {GaN_1_/Al_x_Ga_1x_N_2_/ AlN_40_} (*x* = 0.6)	MO CVD	248		420 702	CNT	8 8	1 3	2″	5.25		2022 Peking Univ.	[122]
20	400 × {GaN_1.5_/AlN_16_}	PA MBE	238-265	15 W 50 W	- -	TE, PC	10 12.5	30 nA 2A	1 µm 4 mm	0.1–0.3		2023, IOFFE and Peking Univ.	[125]

## Data Availability

Data sharing not applicable.

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
