# Peer review of "Electron-Beam-Pumped UVC Emitters Based on an (Al,Ga)N Material System"

_nanomaterials, 2023, doi:10.3390/nano13142080_

Round 1

Reviewer 1 Report

The review paper proposed by Jmerik and co-workers on electron beam pumped UVC emitters is certainly in line with the increasing interest in this wavlength range for sanitation (and other) applications.  My feeling is that the article provides a synthetic introduction to the subject. However, I believe that the paper could be significantly improved by making a couple of mandatory changes as listed below in more details. I also note that the english could be improved and that moderate editing by an english native speaker is necessary before publication.

detailed remarks/comments/suggestions

I) when mentioning, p.3, the "low environmental friendliness" of lamps due to the presence of mercury, it should be useful to mention for completeness that the Minamata Convention, signed by a significant number of UN members has planned to ban the use of mercury-containing devices at short term, making mandatory the development of alternative solutions.

II) An important issue of electron-beam pumping is the radiation damage, related to the life time of devices. In particular, the role of electron bombardement-induced point defect formation should be analyzed in details. My feeling is that a discussion on this issue, which is completely absent in the present manuscript, should be definitely added in order to make the new-comer in the field in position to examine the solution of e-beam pumping in a more balanced way.

III) In the same spirit of providing to the new-comer in the field useful tools to get a comprehensive understanding, I strongly suggest that the authors provide a critical discussion on the parameters affecting the luminescence energy of thin GaN QWs for different AlN barrier thickness. A particularly enlightening paper on this topics has been published by Sun et al, Scientific Reports, 7, 11826 (2017),  (DOI:10.1038/s41598-017-12125-9) and could serve as a base for this discussion. Accordingly, the marked dispersion of the luminescence energy of thin GaN QWs reported in different papers could be more easily understood.

In addition:

1) spelling or grammatical errors are present and should be corrected in line 116, 118, 156, 247, 262, 301, 330, 335, 354, 361, 364, 397, 399, 414, 479, 520, 587-588, 609, 725, 860, 988. as an example. It is likely that this list is not exhaustive

2) the nature of k-sapphire should be identified on first appearance, line 183

3) line 660: what does mean (dots=disks) ??

4) line 663-685: this sentence is confuse and difficult to understand in the present formulation state.

5) line 852: It is not convenient to use the figure numbering of cited papers, which makes the reading confusing

6) The sporadic appearance of cyrillic characters (line 988, No 4 in table 2) should be cleared

7) two figures 17 b are labelled. Please correct.

8) line 620. Where is figure 2c?

moderate editing necessary

Author Response

Dear Editor and Reviewers,

We are thankful to you for your careful reading, as well as for comments and suggestions that eliminate some errors and made our manuscript clearer for readers.

To Reviewer I:

  1. When mentioning, p.3, the "low environmental friendliness" of lamps due to the presence of mercury, it should be useful to mention for completeness that the Minamata Convention, signed by a significant number of UN members has planned to ban the use of mercury-containing devices at short term, making mandatory the development of alternative solutions.
  • We have added in line 132 a reference [37] to the Minamata Convention.
  1. An important issue of electron-beam pumping is the radiation damage, related to the life time of devices. In particular, the role of electron bombardement-induced point defect formation should be analyzed in details. My feeling is that a discussion on this issue, which is completely absent in the present manuscript, should be definitely added in order to make the new-comer in the field in position to examine the solution of e-beam pumping in a more balanced way.
  • We have added in the review section 6 “Preliminary analysis of stability of UVC emitters with e-beam pumping”.
  • In the same spirit of providing to the new-comer in the field useful tools to get a comprehensive understanding, I strongly suggest that the authors provide a critical discussion on the parameters affecting the luminescence energy of thin GaN QWs for different AlN barrier thickness. A particularly enlightening paper on this topics has been published by Sun et al, Scientific Reports, 7, 11826 (2017),  (DOI:10.1038/s41598-017-12125-9) and could serve as a base for this discussion. Accordingly, the marked dispersion of the luminescence energy of thin GaN QWs reported in different papers could be more easily understood.

− We have added in line 1068 a reference [56] to our recent review:
[56]            Jmerik, V.; Toropov, A.; Davydov, V.; Ivanov, S. Monolayer-Thick GaN/AlN Multilayer Heterostructures for Deep-Ultraviolet Optoelectronics, Phys. Status Solidi RRL 2021, 15(9), 2100242. https://doi.org/10.1002/pssr.202100242

where we discussed these issues in detail using the work of Sun et al. and the works of other authors.

In addition:

  • spelling or grammatical errors are present and should be corrected in line 116, 118, 156, 247, 262, 301, 330, 335, 354, 361, 364, 397, 399, 414, 479, 520, 587-588, 609, 725, 860, 988. as an example. It is likely that this list is not exhaustive
  • We tried to address all the issues pointed out by you and correct the manuscript in accordance. All corrections are highlighted in yellow.
  • the nature of k-sapphire should be identified on first appearance, line 183
  • We have omitted the index k- from the text because it is unclear to us too.
  • line 660: what does mean (dots=disks)

− We have replaced this obscure sentence with (quantum dots/disks) and added reference [108] where this is explained.

  • line 663-685: this sentence is confuse and difficult to understand in the present formulation state.

− We have edited this sentence to better understand the contradiction between good optical properties and low output optical power of UVC-LEDs based on ML-thick GaN/AlN MQW structures.

  • line 852: It is not convenient to use the figure numbering of cited papers, which makes the reading confusing

− We have modified this figure and its captions.

  • The sporadic appearance of cyrillic characters (line 988, No 4 in table 2) should be cleared

− We have corrected it.

  • two figures 17 b are labelled. Please correct.

− We have corrected it.

  • line 620. Where is figure 2c?

− This figure is located in page 5, line 39.

Reviewer 2 Report

Although from a scientific point of view, the article is of a high standard, nevertheless it requires some English edition.

Furthermore, few more comments are below:

1.     Line 10. What does the letter C stand for in this abbreviation?

2.     Line 57.  “external electron-(e-) beam”.   Are there internal electron beam ???

3.     Figure 1.  The word “excimer” is much more preferred. For example, in scholar.google.com

https://scholar.google.lv/scholar?hl=en&as_sdt=0%2C5&q=excimer&btnG=

it occurs 507,000 times, while the word “eximer”  is only about 7.690 results

https://scholar.google.lv/scholar?hl=en&as_sdt=0%2C5&q=eximer&btnG=

4.     The paper does not discuss the effects of degradation under the action of electron irradiation. Do point defects form and what role do they play?

5.     To the discussion about carbon nanotubes, it would probably be useful to add information about AlN & GaN nanotubes and their emitting properties under electron irradiation.

Bellucci, S., Popov, A. I., Balasubramanian, C., Cinque, G., Marcelli, A., Karbovnyk, I., ... & Krutyak, N. (2007). Luminescence, vibrational and XANES studies of AlN nanomaterials. Radiation measurements42(4-5), 708-711.

Sekiguchi, T., Hu, J., & Bando, Y. (2004). Cathodoluminescence study of one-dimensional free-standing widegap-semiconductor nanostructures: GaN nanotubes, Si3N4 nanobelts and ZnS/Si nanowires. Journal of electron microscopy53(2), 203-208.

6.     The work somehow lacks the role of ab initio calculations in the analysis and optimization of (Al,Ga)N material system.

7.     Figure 5. This is the original drawing and if not, then a reference is needed.

Author Response

To Reviewer II:

  1. Line 10. What does the letter C stand for in this abbreviation?
  • C is the standard designation for one of the subranges of UV radiation with a wavelength in the range of 100-280 nm. (https://en.wikipedia.org/wiki/Ultraviolet#Subtypes)
  1. Line 57.  “external electron-(e-) beam”.   Are there internal electron beam ???
  • We have deleted this adjective.
  1. Figure 1.  The word “excimer” is much more preferred. For example, in scholar.google.com

https://scholar.google.lv/scholar?hl=en&as_sdt=0%2C5&q=excimer&btnG=it occurs 507,000 times, while the word “eximer”  is only about 7.690 results

https://scholar.google.lv/scholar?hl=en&as_sdt=0%2C5&q=eximer&btnG=

- You are absolutely right. We have corrected this word everywhere.

  1. The paper does not discuss the effects of degradation under the action of electron irradiation. Do point defects form and what role do they play?

− We have added in the review section 6 “Preliminary analysis of stability of UVC emitters with e-beam pumping”.

  1. To the discussion about carbon nanotubes, it would probably be useful to add information about AlN & GaN nanotubes and their emitting properties under electron irradiation.

Bellucci, S., Popov, A. I., Balasubramanian, C., Cinque, G., Marcelli, A., Karbovnyk, I., ... & Krutyak, N. (2007). Luminescence, vibrational and XANES studies of AlN nanomaterials. Radiation measurements, 42(4-5), 708-711.

Sekiguchi, T., Hu, J., & Bando, Y. (2004). Cathodoluminescence study of one-dimensional free-standing widegap-semiconductor nanostructures: GaN nanotubes, Si3N4 nanobelts and ZnS/Si nanowires. Journal of electron microscopy, 53(2), 203-208.

− Thanks for the suggestions, but we cannot include them in the review, in which we consider only structures emitting in the UVC range (<280nm) under electron-beam pumping.

  1. The work somehow lacks the role of ab initio calculations in the analysis and optimization of (Al,Ga)N material system.

− We have added in line 1068 a reference [56] to our recent review:
[56]            Jmerik, V.; Toropov, A.; Davydov, V.; Ivanov, S. Monolayer-Thick GaN/AlN Multilayer Heterostructures for Deep-Ultraviolet Optoelectronics, Phys. Status Solidi RRL 2021, 15(9), 2100242. https://doi.org/10.1002/pssr.202100242

where we discussed these issues in detail.

  1. Figure 5. This is the original drawing and if not, then a reference is needed.

− Figure 5a is the original. Although figure 5b is not original, it may be copied, distributed and/or modified under the terms of the GNU Free Documentation License version 1.2. Subject to the terms of this license, authors may opt out of being named or referenced in derivative works. I will ask the editors for this permission.

Round 2

Reviewer 2 Report

After quite successful revision this manuscript can be recommended for publication.